# DNA punch cards for storing data on native DNA sequences via enzymatic nicking

S. Kasra Tabatabaei[1], Boya Wang[2,8], Nagendra Bala Murali Athreya[3,8], Behnam Enghiad[4], Alvaro Gonzalo Hernandez[5], Christopher J. Fields [6], Jean-Pierre Leburton[3], David Soloveichik[2], Huimin Zhao [1,4,7✉] & Olgica Milenkovic[3✉]

Synthetic DNA-based data storage systems have received significant attention due to the promise of ultrahigh storage density and long-term stability. However, all known platforms suffer from high cost, read-write latency and error-rates that render them noncompetitive with modern storage devices. One means to avoid the above problems is using readily available native DNA. As the sequence content of native DNA is fixed, one can modify the topology instead to encode information. Here, we introduce DNA punch cards, a macromolecular storage mechanism in which data is written in the form of nicks at predetermined positions on the backbone of native double-stranded DNA. The platform accommodates parallel nicking on orthogonal DNA fragments and enzymatic toehold creation that enables single-bit random-access and in-memory computations. We use *Pyrococcus furiosus* Argonaute to punch files into the PCR products of *Escherichia coli* genomic DNA and accurately reconstruct the encoded data through high-throughput sequencing and read alignment.

[1] Center for Biophysics and Quantitative Biology, University of Illinois at Urbana-Champaign, Urbana, IL 61801, USA. [2] Department of Electrical and Computer Engineering, University of Texas at Austin, Austin, TX 78712, USA. [3] Department of Electrical and Computer Engineering, University of Illinois at Urbana-Champaign, Urbana, IL 61801, USA. [4] Department of Chemical and Biomolecular engineering, University of Illinois at Urbana-Champaign, Urbana, IL 61801, USA. [5] Roy J. Carver Biotechnology Center, University of Illinois at Urbana-Champaign, Urbana, IL 61801, USA. [6] High Performance Computing in Biology (HPCBio), Roy J. Carver Biotechnology Center, University of Illinois at Urbana-Champaign, Urbana, IL 61801, USA. [7] Carl R. Woese Institute for Genomic Biology, University of Illinois at Urbana-Champaign, Urbana, IL 61801, USA. [8] These authors contributed equally: Boya Wang, Nagendra Bala Murali Athreya. ✉email: zhao5@illinois.edu; milenkov@illinois.edu

All existing DNA-based data recording architectures store user content in synthetic DNA oligos[1–12] and retrieve desired information via next-generation (NGS; HiSeq and MiSeq) or third generation nanopore sequencing technologies[6]. Although DNA sequencing can be performed routinely and at low cost, de novo synthesis of DNA strands with a predetermined nucleotide content is a major bottleneck due to multiple issues[13]. First, DNA synthesis protocols add one nucleotide per cycle, with each cycle taking seconds, and therefore are inherently slow and prohibitively expensive compared to existing optical and magnetic writing mechanisms. Second, DNA synthesis is an error-prone procedure that often fails to account for a large volume of information-bearing oligos, and which leads to oligo pools that contain substitution and indel errors[14]. Third, the available mass per synthetic DNA oligo is usually small, enabling only a limited number of readouts[9].

To address the above limitations of DNA-based data storage systems and pave the way towards future low-cost molecular storage solutions we propose a storage paradigm that represents information via in vitro topological modifications on native DNA (e.g., genomic DNA, cloned, or PCR-amplified products). Unlike all previously proposed methods for DNA-based data storage, our system stores information in the sugar-phosphate backbone of DNA molecules rather than their sequence content. More precisely, binary information-bearing strings are converted into positional encodings that describe if a carefully preselected set of nicking sites is to be nicked or not. The information stored in nicks can be retrieved in an error-free manner using NGS technologies, similar to synthesis-based approaches. This is accomplished through alignment of DNA fragments obtained through the nicking process to the known reference genomic DNA strands. Due to the availability of the reference even very small fragment coverages lead to error-free readouts, which is an important feature of the system. Alternative readout approaches, such as non-destructive solid-state nanopore sequencing, can be used instead, provided that further advancement of the related technologies enable high readout precision.

Nick-based storage also allows for introducing a number of additional functionalities into the storage system, such as bitwise random access and pooling—both reported in this work—and in-memory computing solutions reported in a follow-up paper[15]. These features come at the cost of reduced storage density which is roughly 50-fold smaller than that of synthetic DNA-based platforms.

It is important to point out that although there are many different means for topologically modifying DNA molecules, for data storage applications enzymatic nicking (i.e., creating a single-bond cut in the DNA sugar-phosphate backbone via an enzyme) appears to be the most specific, efficient, and versatile recording method. DNA methylation, a well-known type of modification that may be imposed on cytosine bases, is insufficiently versatile and hard to impose in a selective manner. Topological alterations in DNA in the form of secondary structure may not be cost- or density-efficient as additional nucleotides are needed to create such structures[16].

The proposed native DNA-based storage architecture is depicted in Fig. 1, encompassing a Write and Read unit. The Write unit (Fig. 1, Top) involves extracting native DNA registers

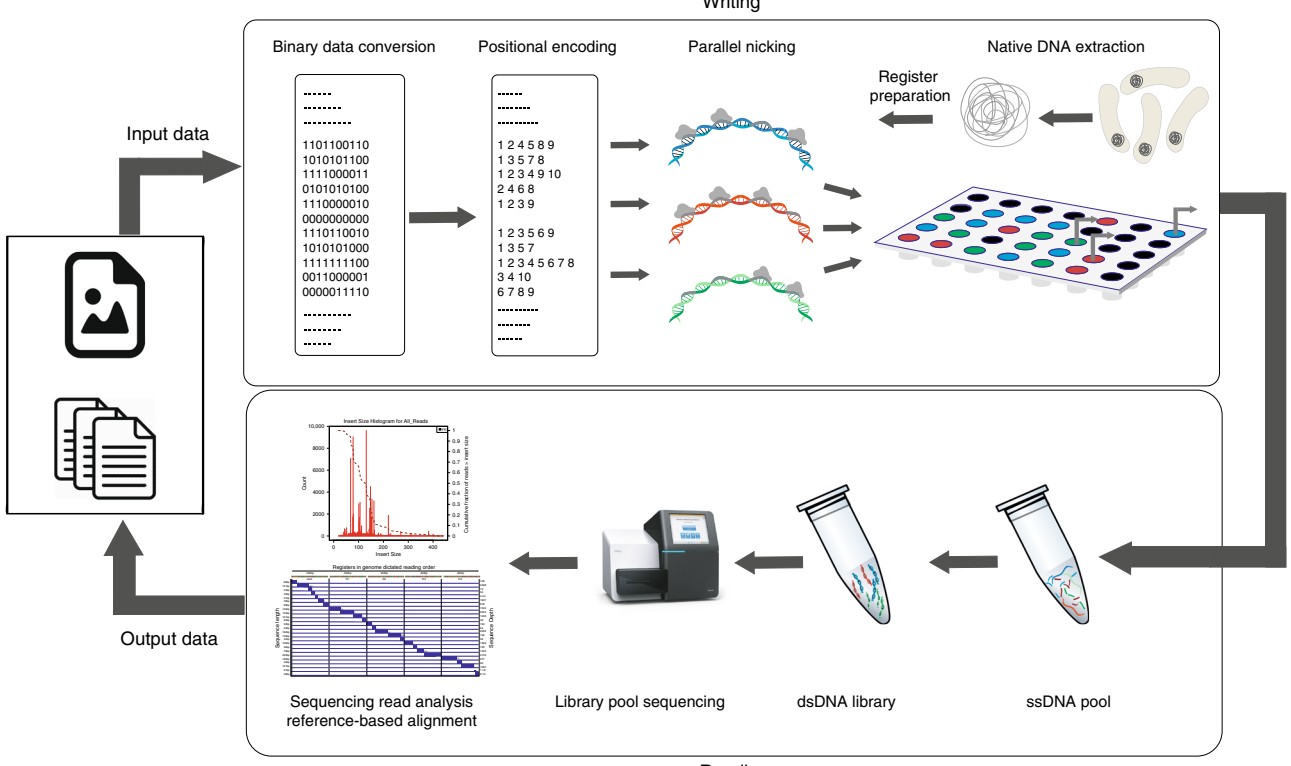

**Fig. 1 The native DNA-based data storage platform.** In the Write component, arbitrary user content is converted into a binary message. The message is then parsed into blocks of $m$ bits, where $m$ corresponds to the number of nicking positions on the register (for the running example, $m = 10$). Different (orthogonal) registers may be used to encode information in parallel, as indicated by the different colors of the DNA strings. Subsequently, binary information is translated into positional information indicating where to nick. Nicking reactions are performed in parallel via combinations of PfAgo and guides. In the Read component, nicked products are purified and denatured to obtain a pool of ssDNAs of different lengths. The pool of ssDNAs is sequenced via MiSeq either as part of an orthogonal register mixture or individually. The output reads are processed by first performing reference-based alignment of the reads, and then using read coverages to determine the nicked positions.

to be used as recording media, identifying proper nicking sites and encoding the data through positional codes. It also includes a parallel nicking procedure centered around highly specific controllable nicking endonucleases. The Read unit (Fig. 1, Bottom) involves dsDNA denaturation, library preparation, and NGS sequencing followed by read alignment and decoding.

The individual components of the Write and Read system are described in detail in the Results section.

## Results

**The write architecture.** To implement a nick-based data storage platform, one first needs to identify a nicking enzyme with optimized programmability. Until this work, this remained a challenge for reasons described in what follows.

Known nickases (natural/engineered) are only able to detect and bind specific sequences in DNA strands that tend to be highly restricted by their context. For example, nicking endonucleases only recognize specific sequences, usually 6 bps long, which makes them difficult to use for large-scale recording. Similarly, *Streptococcus pyogenes* Cas9 nickase (*Sp*Cas9n), a widely used tool for genetic engineering applications, requires the presence of a protospacer adjacent motif (PAM) sequence (NGG) at the 3′ site of the target DNA. The NGG motif constraint limits the nicking space to 1/16 of the available positions due to the GG dinucleotide in the PAM sequence. The *Sp*Cas9n complex uses RNA guides (gRNAs) to bind the target which makes it rather unstable and hard to handle. Furthermore, *Sp*Cas9n is a single turnover enzyme[17], i.e., one molecule of the enzyme can generate only one nick in a DNA molecule. These properties make *Sp*Cas9n exhibit low efficiency and versatility for storage applications.

To address these issues, we propose to use the DNA-guided programmable restriction enzyme *Pyrococcus furiosus* Argonaute (*Pf*Ago)[18] as our writing tool. *Pf*Ago is a highly accurate artificial restriction enzyme (ARE) for efficient double-stranded cleavage at arbitrary sites that generates defined sticky ends of various lengths[18]. The enzyme has a significantly larger flexibility in double-stranded DNA cleaving compared to the Cas9 nickase as it is not restricted by the presence of any special sequence at its recognition site. Most importantly, the enzyme has a high turnover rate as one enzyme molecule can be used to create a large number (~hundreds) of nicks. *Pf*Ago also uses 16 nt DNA guides (gDNAs) that are stable, cheap to acquire and easy to handle in vitro. *Pf*Ago has found various applications in genetic engineering and genomic studies, such as DNA assembly and DNA fingerprinting[18]. DNA cleavage using this enzyme requires the presence of two gDNAs, targeting both strands at positions close to each other, which is not needed nor desirable for storage applications.

We hence performed extensive experimental studies to demonstrate that under proper reaction conditions (e.g., buffer and temperature), *Pf*Ago with single gDNAs can also target only one of the DNA strands and successfully perform simultaneous nicking of multiple prescribed sites on that strand with high efficiency and precision within 40 min. Supplementary Figs. 1, 2 illustrate the process of verifying the activity of the *Pf*Ago's nicking enzyme. Also, a comparative analysis of the nicking performance of *Sp*Cas9n, another potential DNA nicking enzyme, and *Pf*Ago is provided in the Supplementary Table 1 and Supplementary Figs. 3–4.

DNA register and nicking site selection: The genomic DNA was extracted from a culture of *E. coli* K12 MG1655 grown overnight. We found 450 bps to be a suitable length for our recording DNA fragments, henceforth referred to as registers, given that:

The length should accommodate a sufficiently large number of nicking sites. The working example register used for file storage includes ten sites, which are separated by at least 25 bps. This distance constraint is imposed to ensure that inter-nick strands do not disassociate at room temperature and that the intra-nicking fragments are of sufficiently long length. Other length choices are possible as long as they do not lead to undesirable strand disassociations.

As the nicking registers have to include nicking recognition sites that have large edit distances so as to avoid nonspecific nicking, one long content-predefined native DNA register may not be as versatile as a collection of shorter registers selected from different parts of the genomic sequence.

The length should be adequate for both NGS platforms and solid-state nanopores reading. Also, the selected length makes post-sequencing analysis and reference alignments easier and less computationally demanding.

Therefore, several 450 bp candidate registers were PCR amplified from the *E. coli* genomic DNA. Of these candidates, five registers were selected for system implementation and testing. The registers contain between five and ten nicking sites, all located on the antisense (bottom) strand (Supplementary Figs. 5a, 9b–e). The sequences of length 16 corresponding to the nicking sites are selected to be at a Hamming distance >8 in order to avoid nonspecific nicking. Supplementary Table 2 list the count of distinct $k$-mers ($k = 1, 2, …, 13$) found in *E.coli* genome. As the enzyme cuts a certain site with high accuracy (between positions 10 and 11 of the corresponding gDNA[18]), this selection eases the way towards predicting the lengths that are supposed be observed after sequencing. As a result, each register is associated with a particular set of intra-nicking fragment lengths that are expected to arise upon the completion of the nicking reactions.

Positional encoding: Each register contains $n$ designated nicking positions. As previously described, even though long registers of length up to few kbs can be easily accommodated and amplified via PCR, multiple registers are preferred to long registers. The nicking positions at each strand are determined based on four straightforward to accommodate sequence composition constraints described in the Supplementary Information, Section B.1.

To perform the actual encoding, user files are parsed into n-bit strings which are converted into nicking positions of spatially arranged registers, according to the rule that a 1 corresponds to a nick while a 0 corresponds to the absence of a nick. The number of bits recorded is chosen based on the density of nicks and the length of the register. As an example, the string 0110000100 is converted into the positional code 238, indicating that nicking needs to be performed at the 2nd, 3rd, and 8th positions (Fig. 2a). Note that recording the bit 0 does not require any reactions, as it corresponds to the no nick ground state. Therefore, nick-based recording effectively reduces the size of the file to be actually recorded by half. This property of native DNA storage resembles that of compact disks (CD) and other recorders. Note that the length, sequence composition and nicking sites of a register are all known beforehand, so that reading amounts to detecting the positions of the nicks.

Data recording: In the write component of the proposed system, binary user information is converted into a positional code that describes where native DNA sequence is to be topologically modified, i.e., nicked. Each nick encodes either $log_2 2 = 1$ bit (if only one strand is allowed to be nicked or left unchanged) or $log_2 3 = 1.58$ bits (if either of the two strands is allowed to be nicked or both left unchanged). Each register is nicked by mixing it with a combination of nicking enzymes that contain guides matched to the collection of sites to be nicked and

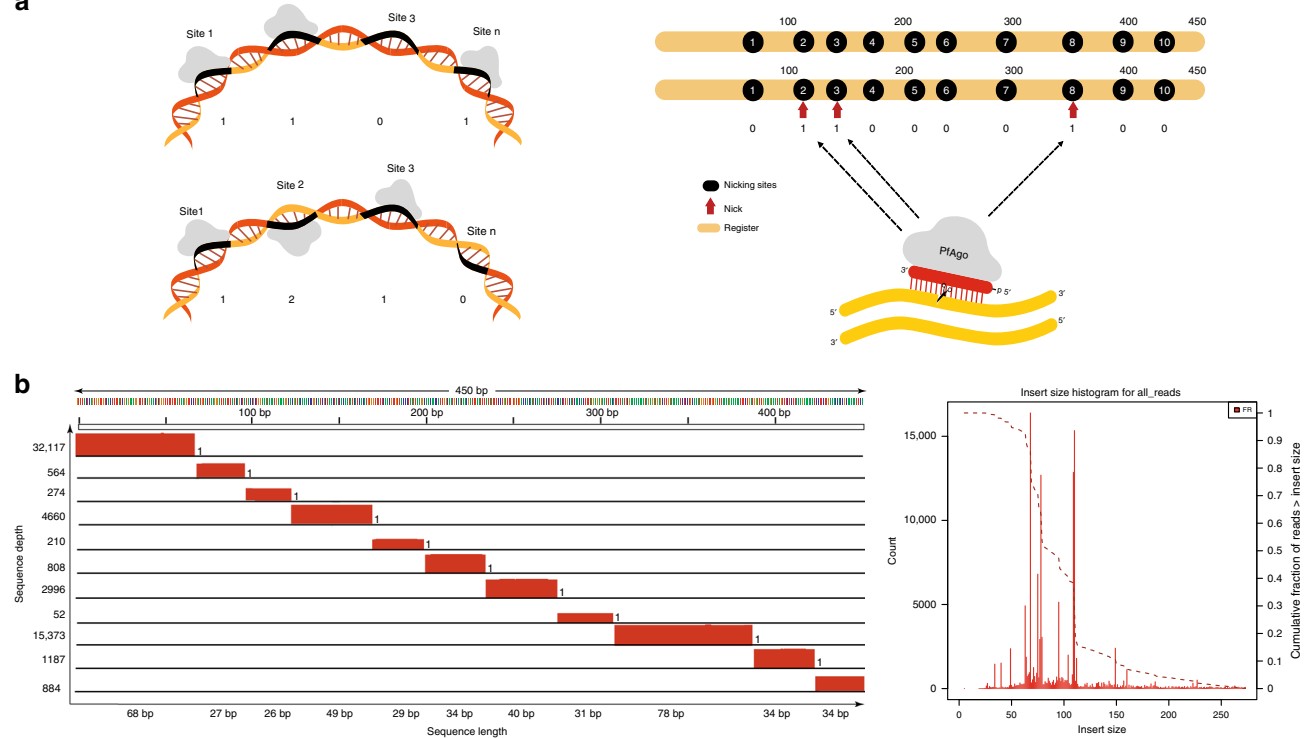

**Fig. 2 Writing and reading the encoded data. a** *Pf*Ago can nick several pre-designated locations on only one strand (left, top) or both strands (left, bottom), simultaneously. In the first register, the stored content is 110...1, while in the second register, the content is 121...0. The chosen register is a PCR product of a 450 bp *E. coli* genomic DNA fragment with 10 pre-designated non-uniformly spaced nicking positions. The positional code 238 corresponds to the binary vector 0110000100 (right). **b** The MiSeq sequencing reads were aligned to the reference register to determine the positions of the nicks. The size distribution histogram (right) and coverage plots (left) are then generated based on the frequency and coverage depth of the reads. Coverage plots allow for straightforward detection of nicked and unnicked sites. In the example shown, all the ten positions were nicked, resulting in eleven aligned fragments. Source data are available in the Source Data file.

the information content to be stored. Nicking is performed in parallel on all sites and the recording process takes ~40 min.

To enable fast and efficient data recording, a library of registers with desired nicking site patterns can be created in a combinatorial fashion and in parallel using automated liquid handlers. To this end, we designed *Pf*Ago guides for all predetermined nicking positions in the chosen register and created registers bearing all $2^{10} = 1024$ nicking combinations (Supplementary Table 3). The recording protocols for the registers are described in the Supplementary Information, Section B.4.

Data organization: The registers or mixtures of registers are placed into grids of microplates that enable random access to registers and spatially organize the data, similar to tracks and sectors on disks and tapes. The placement is dictated by the content to be encoded. For our proposed topological storage system prototype, the plate-based solution facilitates specific access to data both based on their well position and based on their content, through the use of toehold bitwise random access described in the Additional features section. The proposed positional organization also limits the scope/extent of data destruction during reading, as information is stored in a distributed manner and random-access methods based on PCR may only skew the content in a small number of wells. In its current form, the well-plate approach is not scalable and does not offer the desired storage densities. Nevertheless, there are several related efforts underway for designing scalable positional data storage systems using MALDI plates[19] or microfluidic devices[20]. These solutions are currently not able to handle large data volumes but are expected to improve significantly in the near future.

**The read architecture**. The nicked registers are first denatured, resulting in ssDNA fragments of variable lengths dictated by the nicked positions. These length-modulated ssDNA fragments are subsequently converted into a dsDNA library, sequenced on Illumina MiSeq, and the resulting reads are aligned to the known reference register sequence. The positions of the nicks are determined based on read coverage analysis, the insert size distributions and through alignment with the reference sequence; potential nicking sites that are not covered are declared to be 0s (Fig. 2a, b).

The NGS sequencing process: Since a nick is a very small alteration in the DNA backbone, direct detection of this topological modification on a long DNA strand remains challenging. However, if one denatures the dsDNA and sequences the resulting intra-nick fragments, nick reading becomes straightforward. Hence, in the read component of our system, nicked DNA is first converted to single strands at high temperature; the resulting pool of ssDNA is converted into a library of dsDNA and this library is sequenced using a MiSeq device. The positions of the nicks are determined via read analysis and subsequent reference-based sequence alignment, as the native DNA registers are known a priori.

Since one needs to sequence the whole fragments (in our case 450 bp) to retrieve the positions of the nicks, the readout yield is reduced at least one order of magnitude. However, it is important to note that in our system, high coverages are not essential for accurate data readouts, as even relatively small coverages (as low as one read) can be resolved without errors given that the reads can always be aligned to the reference. Also, as the fragment lengths are co-dependent and as the fragments have to jointly

tile/cover the known reference DNA string, certain missing fragments can be accommodated as well. In simple terms, if one fragment has low or no coverage but leaves a hole in the aligned strings with respect to the reference, its presence is noted indirectly upon alignment.

NGS readout and alignment experiments: As a proof of concept, we report write-read results for two compressed files, containing a 272-word text file of size 0.4 KB (originally of size 2 KB) containing Lincoln's Gettysburg Address (LGA) and a JPEG image of the Lincoln Memorial of size 14 KB (originally of size 329 KB). Both files were converted into ASCII format and retrieved with perfect accuracy. Supplementary Fig. 7 shows the distribution of all 10-bit substrings in the files.

Of the whole stored data, a randomly selected set of samples encoding 80 bits, including $5 \times 10$-bit registers from the image and $3 \times 10$-bit registers from the text file, was sequenced. Given the inherent redundancy of the sequencing process and the careful selection of the nicking sites and register sequences, no error-correction redundancy was needed, and data was perfectly retrieved. As may be observed from Figs. 2, 3 and Supplementary Figs. 5, 6, peak coverages are in perfect agreement with the fragment lengths predicted based on the nicking location.

**Additional functionalities.** In order to increase the per register and per-well storage density of the system we also introduced and experimentally tested three additional nick-based storage methods. The first method includes mixing what we refer to as orthogonal registers: Orthogonal registers are DNA fragments isolated from different parts of the genomic DNA that have small sequence similarity scores. The second method involves nicking DNA both at the sense and antisense strands in which case one achieves a 58% increase in density. The third method involves combinatorial mixing of the same register as dictated by a mathematically grounded group testing scheme (discussed in the section "Combinatorial mixing").

Multi-register recording: To prove the principle of orthogonal register recording, we use the running example LGA register as well as four other orthogonal registers each of 450 bp length. All registers are extracted from the *E. coli* genome (O1–O4), they do not share substrings used as nicking sites and have <50% pairwise similarity matches. Furthermore, the bits stored in the registers are naturally ordered as the registers have an inherent order within their native genomic string. Using orthogonal registers, we designed a total number of 32 nicking positions on all the registers and encoded the title Gettysburg Address of size 126 bits via simultaneous nicking of all registers mixed together. This piece of data was also successfully recalled without errors as shown in (Fig. 3a).

Sense–antisense recording: As a proof of concept, we used the running example DNA register (LGA) to test recording on both sense and antisense strands. In this setting, the recording alphabet is ternary, i.e., at any predesignated nicking position, a nick on the sense strand denotes 1, a nick on the antisense denotes 2, and no nick denotes 0. This increases the potential storable data for every nicking site ~1.58 fold. We also experimentally showed that our register can be nicked on 10 prescribed sites on both strands, six of which are located on the sense strand and four on the antisense

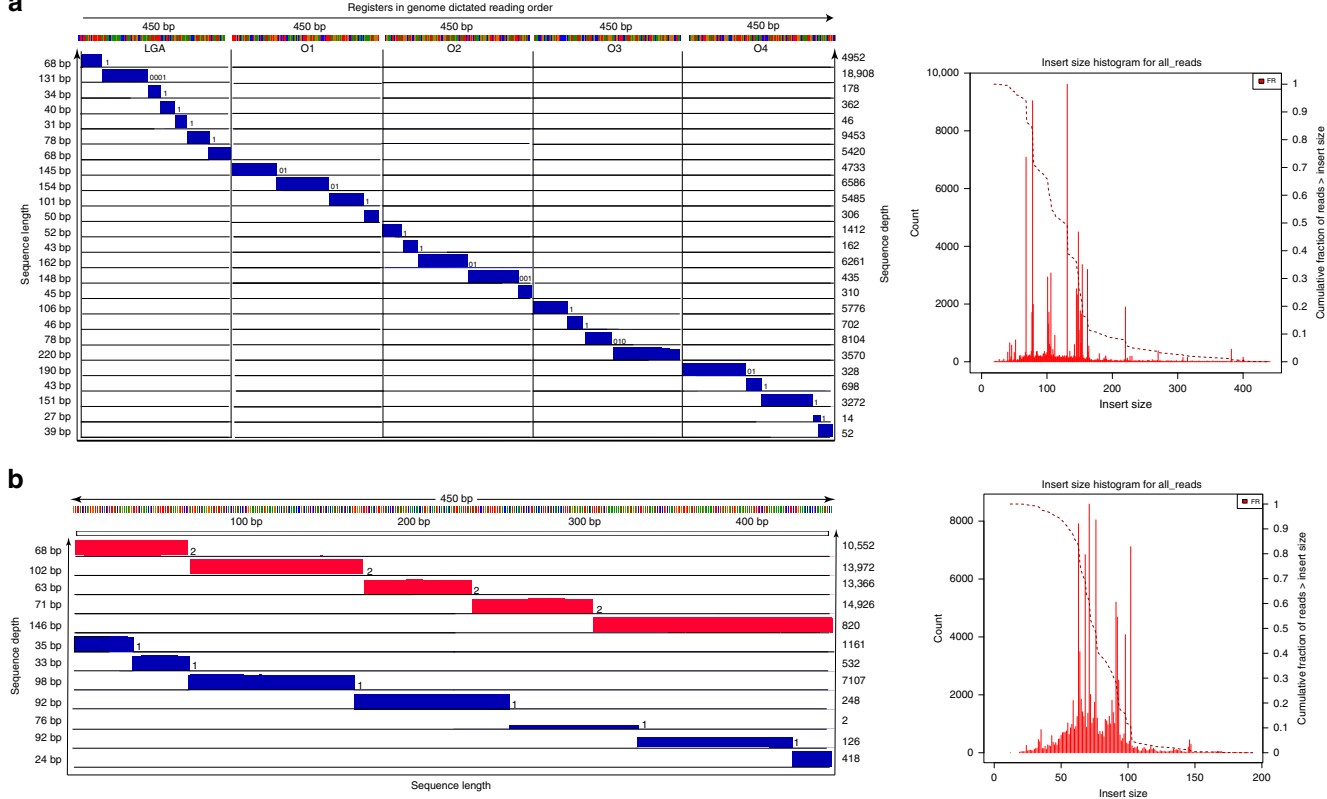

**Fig. 3 Multi-register and sense-antisense recording. a** Five orthogonal registers used instead of one single register. Each vertical section represents one register in genome dictated reading order, and each row shows the read lengths retrieved after sequencing analysis. Read lengths are recorded on the left and sequencing depths on the right axis. **b** Nicking both the sense and antisense strands. Here, the binary format is switched to ternary format, i.e., for each nicking site, a nick on the sense strand denotes 1, a nick on the antisense denotes 2, and no nick denotes 0. The coverage (left) and size distribution (right) plots verify that the register can be nicked on ten prescribed sites on both strands, six of which are located on the sense strand and four on the antisense strands, corresponding to the 10-bit string of data: 1212121211. The sense-antisense nicking approach may also be used for parallel recording on multiple registers. Source data are available in the Source Data file.

strands, corresponding to the 10-bit string of data: 1212121211 (Fig. 3b).

Other technical details regarding implementations with orthogonal registers and with nicks on both DNA strands are provided in the Supplementary Information, Section B.6 and Supplementary Figs. 8–11.

Alternative readout platforms: A faster, portable and more cost-effective method for reading the nicked DNA registers is via two-dimensional (2D) solid-state nanopore membranes[21]. Solid state nanopores also enable reading single information-bearing DNA molecules nondestructively and hence increase the lifecycle of molecular storage systems.

The main drawback of existing solid-state nanopore technologies is the lack of translocation controls that ensure more precise and less noisy nick-position readouts. One approach to mitigate this problem, described in the ref. [21], is to extend nicks to toehold, short single-stranded regions on dsDNA created through two closely placed nicks, instead of single nicks. As shown in the Supplementary Information, Section B.8., *Pf*Ago can be used to create toeholds of arbitrary lengths. Experimental evidence[21] reveals that toeholds can be easily and accurately detected using solid-state $SiN_x$ and $MoS_2$ nanopores.

However, the cost of creating toeholds is twice as high as that of nicks, since one needs two different nicking guides. Hence, one needs to identify alternative mechanisms for direct nick detection. To illustrate the feasibility of the direct readout approach, we performed molecular dynamics (MD) simulations along with quantum transport calculations to obtain the ionic and transverse current signals. In order to reduce the computational time of the MD simulations, DNA strands of 30 nucleotides with only one nicked site in the middle were considered. The simulations were performed at 1 V, with a total translocation time of a few nanoseconds. The results indicate a strong inverse correlation between the ionic and electronic sheet current signals along the membrane induced by nicks in $MoS_2$ nanopores (Supplementary Figs. 12–14 and Supplementary Movie 1). We also provide an additional analysis in support of our findings by calculating the Pearson's correlation between the normalized signals to indicate the time frame when the nick is residing within the pore (Supplementary Fig. 12c).

Bitwise random access: In addition to allowing for nanopore-based readouts, toeholds may be used for bitwise random access and toehold-mediated DNA strand displacement, used for engineering dynamic molecular systems and performing molecular computations[22–24]. Information is processed through releasing strands in a controlled fashion, with toeholds serving as initiation sites to which these input strands bind to displace a previously bound output strand.

Toeholds are usually generated by binding two regions of synthetic ssDNA and leaving a short fragment unbound. However, we experimentally demonstrated that *Pf*Ago can easily create toeholds in native DNA. To form a toehold, we introduce two nicks at a small distance from each other (in our experiment, 14 bps). Under appropriate buffer and temperature conditions, in a single reaction the strand between the two nicks disassociates, leaving a toehold on the double-stranded DNA (Supplementary Fig. 15).

Fluorescence-based toehold-reporting methods can be used to detect toeholds in a nondestructive manner and hence enable symbol-wise random access. In addition, these methods may be used to estimate the concentration of registers bearing a toehold without modifying the DNA registers. We illustrate this process on a register encoding 0010000000, with a toehold of length 14 nts at the nicking position 3 (Supplementary Fig. 16). As shown in Fig. 4a, a fluorophore and quencher labeled reporter strand with a sequence complementary to the toehold can hybridize to the toehold segment, producing a fluorescence signal resulting from an increase of the distance between the fluorophore and the quencher. We were also able to reliably measure different ratios of native DNA fragments with and without toeholds within 20 min (Fig. 4b). Since the reporter has a short single stranded overhang, it can be pulled off from the register upon hybridization, making the readout process non-destructive (Polyacrylamide gel electrophoresis analysis, Fig. 4c). This feature equips our proposed storage system with unique nondestructive bitwise random access, since one is able to design specific reporters to detect any desired toehold sequence which accompanies a nick. It also enables in-memory computations on data encoded in nicks as demonstrated in the ref. [15].

Note that since the positions of the nicks are predetermined and since orthogonal registers are mixed, one only needs to have one reporter sequence available for each bit location. For LGA register, one need only ten different fluorophores for bitwise random access to all ten positions simultaneously. If access is sequential, only one fluorophore is needed. This is to be contrasted with the first random access scheme we proposed and reported on in the ref. [4]. There, one can only access individual oligos through a time-consuming process of PCR and sequencing.

Combinatorial mixing: As each register sequence requires different nicking guides to be used with the nicking endonuclease, the question arises if one can store information on multiple copies of the same register bearing different nicking patterns. Since there is no inherent ordering for the information stored on the same register string within a mixed pool, NGS sequencers cannot resolve the origin of the different fragments through alignment. Several solutions for this problem are possible. One solution is to store information in the number of nicks, rather than their locations. Another, more effective approach is to use a combinatorial mixing scheme such as group testing or adder channel coding. Group testing effectively allows for mixing of $k$-out-of-$N$ registers, where $k$ is significantly smaller than $N$ and the number of bits scales as $k\log_2 N$. The optimal mixing scheme for adder channels is the Lindström scheme[25], which does not impose any restrictions on the value of $k$.

In a nutshell, not all combinations of nicking patterns on the same register sequence can be mixed as one has to prevent multiple distinct alignment solutions for the union of the fragments generated from the mixture. This is accomplished through careful implementation of the Lindström scheme. This coding scheme was originally designed to allow for identifying individual components in a real-valued sum of binary weighted vectors. The scheme is asymptotically optimal in so far that it can discriminate between any of the $2^N$ distinct sums of $N$ binary vectors of length as small as $M = \frac{2N}{\log_2 N}$. The mathematical explanation behind the scheme is described in detail in Section B.11 of the Supplementary Information, along with a number of experimental test results that indicate that the underlying combinatorial mixtures can be sequenced together and reassembled without errors (Supplementary Fig. 17).

## Discussion

By reprogramming *Pf*Ago to serve as a universal nickase and using native *E. coli* DNA strings we implemented a DNA-based storage system that mitigates the use of long synthetic DNA strands for storing user information, and records data in DNA backbone rather than the sequence content. In contrast to synthesis-based methods, our platform has significantly reduced writing latency due to the ability of the enzyme to encode information in parallel. Also, our approach has exceptionally high reliability, in contrast to synthetic DNA-based data storage that

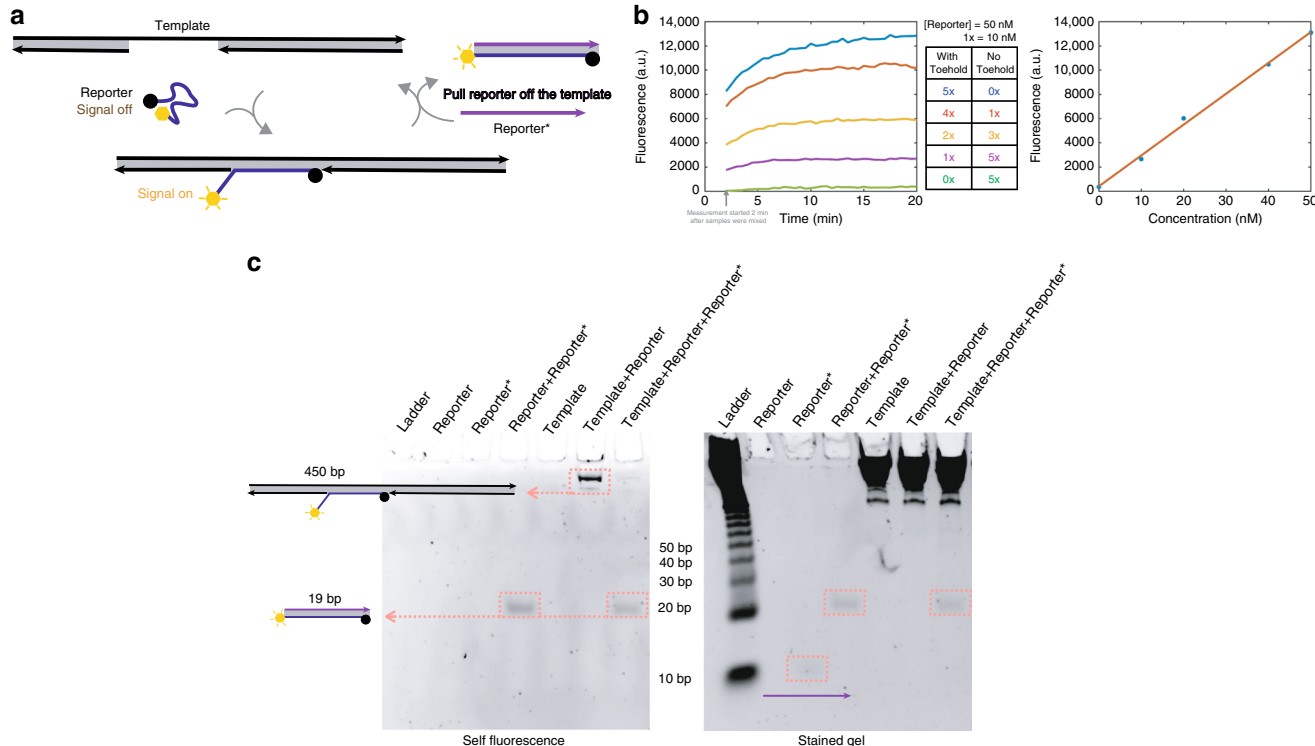

**Fig. 4 Non-destructive bitwise random access. a** Non-destructive detection of toeholds through a fluorophore and quencher labeled Reporter strand. Once the Reporter hybridizes with the toehold on the register strand, a fluorescence signal is observed due to the increase of the distance between the fluorophore and quencher. The Reporter strand can be pulled off from the register once the Reporter* strand hybridizes with the Reporter. **b** Kinetics of detecting the concentrations of registers with and without toeholds in a mixtures (left). The fluorescence signals saturate within 20 min. The samples were mixed no more than 2 min before measurement. The concentration of toehold-ed DNA can be accurately quantified through fluorescence intensity (right), as it increases linearly with the concentration of the registers with toehold. **c** PAGE gel results for non-destructive detection of a toehold. Left: the gel was not stained with other fluorescence dyes and hence only the species with self-fluorescence are observed. After adding the Reporter, a large size complex appears in the lane "Template+Reporter", indicating hybridization of the Reporter and the register. After the Reporter* is added, as seen in the lane "Template+Reporter+Reporter*", the large size complex in the previous lane no longer exhibits self-fluorescence, indicating that the Reporter strand is pulled off from the register. Right: The same gel after staining (with SYBR gold: ThermoFisher Scientific). Note that the same molecular components are observed, in addition to the Reporter*. The PAGE analysis was performed once as this is a standard molecular computing setup and the results are expected to be consistent over many trials. Source data is available in the Source Data file.

suffer from high rates of synthesis and missing oligo errors (Table 1). Note that the inherently high error-rate of synthetic platforms is further amplified when storing compressed data since errors in compressed formats tend to exhibit catastrophic error propagation. Hence, nick-based storage schemes are exceptionally well-suited for storing compressed data representation. Furthermore, as nicks may be easily superimposed on synthetic DNA molecules, they may also be used to encode metadata that can be erased (through simple ligation) and recreated or altered in a time efficient manner.

Our DNA punch card storage system design also enables enzyme driven toehold creation for bitwise random access, a unique feature not paralleled by any other implementation. Furthermore, nick-based storage allows for in-memory computing on the data stored on molecular media (Table 1). As reported in a follow-up paper[15], DNA-based strand displacement data processing schemes capable of parallel, in-memory computation, can perform on such system and thereby eliminate the need for sequencing and synthesizing new DNA on each data update.

With further improvements and optimization, our system may also allow for cost-efficient scaling since a. long registers and mixtures of orthogonal registers may be nicked simultaneously; b. most compressed data files do not contain all possible 10-mers or compositions of orthogonal *k*-mers so that not all guide

combinations are needed. c. genomic DNA and *Pf*Ago, as the writing tool, are easily extracted and used in standard laboratory settings, and the mass of the created DNA products by far exceeds that of synthetic DNA. This may significantly increase the number of readout cycles with NGS devices.

## Materials and methods
**Guide DNA selection and positional coding**. To minimize off-site nicking rates, increase the efficiency and accuracy of gDNA-binding, and eliminate readout errors, the nicking regions were selected by imposing the following constraints: each region is of length 16 bps, which allows each gDNA to bind to a unique position in the chosen registers; each region has a GC content in the range 20–60%; there are no G repeats longer than three (this feature is only needed in conjunction with nanopore sequencing); the gDNAs are at Hamming distance at least eight from each other; and the nicking sites are placed at least 25 bps apart. Positional coding is performed based on the number of orthogonal registers used, and the number of nicking positions selected on each register. For the single-register implementation with one-sided nicking, ten positions were selected on a 450 bp genomic fragment, bearing ten bits.

Although choosing longer registers with a larger number of nicking positions is possible and indeed easily doable, we selected the given length for our proofs of concept in order to accommodate different sequencing technologies. The five orthogonal register implementation may encode 32 bits with one sided nicking, and roughly 50 bits with two-sided nicking. Hence, each binary encoded message is parsed into blocks of length either 10, or 32 or 50 bits, which are recorded via nicking. The sequences of all the registers are provided in Supplementary Table 4. Supplementary Table 5 lists all the gDNAs used for nicking, along with other oligos used in this work (primers and reporters).

**Table 1 Comparison of synthetic and native DNA-based data storage platforms.**

| DNA-based storage method | Writing latency | Reading latency | Enables computation? | Bit-wise random access | Maximum achievable physical density | Information density | (Optimal) coding loss[10] |
|---|---|---|---|---|---|---|---|
| Synthesis-based | Sequential de novo synthesis/ hours | NGS/hours | × | × | 200 Ebytes/g[9] | <2 bits/bp (to account for coding loss, usually ~1.5 bits/bp) | 21%[10,11] |
| This work | Parallel nicking/ <40 min | NGS followed by reference alignment/ hours | ✓□ | ✓□ | 4 Ebytes/g | 0.036 bits/bp | 0% |

Native DNA-based platforms outperform synthetic DNA-based approaches in all performance categories, except for storage density.

**Genomic DNA isolation and PCR amplification**. Genomic DNA was extracted from an overnight culture of *E. coli* K12 MG1655, using the Wizard® Genomic DNA Purification Kit (Promega). The kit can be used for at least 100 isolations. One extraction yields up to 100 μg of genomic DNA (from 5 ml overnight culture) which can be used for several hundreds of amplification reactions. Isolated genomic DNA is subsequently stored at 4 °C. Two means of reducing the cost of DNA extraction are to either manually extract the DNA or fully automate the process. The former approach can lead to a very high yield of DNA at a small cost, and all used buffers and reagents can be made in-house. For more information, see https://bio-protocol.org/bio101/e97#biaoti1286.

DNA amplification was performed via PCR using the Q5 DNA polymerase and 5× Q5 buffer (New England Biolabs) in 50 μl. All primers purchased from Integrated DNA Technologies (IDT). In all PCR reactions, 10–50 ng of *E. coli* genomic DNA and 25 pmol of forward and reverse primers were used. The PCR protocol consists of: (1) 3 min at 98 °C, (2) 20 s at 98 °C, (3) 20 s at 62 °C, (4) 15 s at 72 °C, (5) go to step 2 and repeat the cycle 36 times, (6) 8 min at 72 °C. Each PCR reaction produced ~2–2.5 μg of the register string, sufficient for >100 reactions. PCR products were run on 1% agarose gel and purified using the Zymoclean gel DNA recovery kit (Zymo Research). Supplementary Table 4 lists all the PCR products used in this work. Supplementary Table 5 contains the primers used for amplifying these sequences.

**Enzyme expression and purification**. Enzyme expression and purification was performed as described in the ref. [18] with some modifications: A strep(II)-tagged (N-terminal) codon-optimized version of *Pf*Ago gene was ordered from GeneScript and cloned into pET28a plasmid to yield the expression plasmid pHZ-*Pf*Ago. The expression plasmid was transformed into *Escherichia coli* KRX (Promega) according to manufacturer's protocol. The strain was cultivated in LB medium supplemented with 0.4% (w/v) glucose and 50 μg/mL kanamycin at 37 °C. After the incubation overnight, the culture was centrifuged at 3220 × *g* for 5 min and the supernatant was removed. The pellet was then resuspended in Terrific Broth containing 50 μg/mL kanamycin and incubated at 37 °C until the $OD_{600}$ = 1.2–1.5. The culture was cold-shocked in an ice bath for 15 min. *Pf*Ago expression was induced by adding isopropyl-b-D-thiogalactoside (IPTG) and L-Rhamnose to a final concentration of 1 mM and 0.1% (w/v). The expression was continued by incubation at 30 °C for ~20 h.

Cells were collected by centrifugation for 15 min at 6000 × *g*, and frozen afterwards. After thawing, they were resuspended in Buffer A (1 M NaCl, 20 mM Tris/HCl pH = 8.0) and disrupted by sonication with 20 30 s pulses at 30% power with 30 s pause in between. The solution was centrifuged for 30 min at high speed at 4 °C, and the supernatant was used for purification by strep-tactin superflow high capacity cartridge (IBA, Germany). Before loading the supernatant, the column was equilibrated with Buffer A. After loading, the column was washed with 10 CV (column volumes) Buffer A. N-terminally Strep (II)-tagged *Pf*Ago was eluted in Buffer B (1 M NaCl, 20 mM Tris/HCl pH = 8.0, 2.5 mM biotin). The protein was stored in buffer C (300 mM NaCl, 20 mM Tris–HCl, pH = 8.0, 15% (v/v) glycerol) and the aliquots were stored at −80 °C. More than 200 nmols of *Pf*Ago was purified from 1 L of *E. coli* culture, enabling >50,000 reactions.

**PfAgo nicking experiments**. For ease of access and spatial organization of data, pools of registers are kept in 384-well plates. The distribution of the registers and enzymatic reagents was performed manually (when transferring volumes of reagents with volumes of the order of microliters) or using the Echo® 550 liquid handler (LABCYTE) (when transferring volumes of reagents of the order of nanoliters). The latter allows for testing the reaction efficiency of nanoliters of reagents and it also represents a faster and more efficient manner of liquid handling at larger scales. *Pf*Ago reactions were performed in buffer conditions including 2 mM $MnCl_2$ 150 mM NaCl, and 20 mM HEPES, pH 7.5, and a total volume of 10–50 μL. After adding the buffer, dsDNA registers, ssDNA

phosphorylated gDNAs and the enzyme, the sample was thoroughly mixed by pipetting 6–8 times. Nicking was performed based on the following protocol: (1) 15 min at 70 °C, (2) 10 min at 95 °C, (3) gradual decrease of temperature (0.1 °C/s) to 4 °C. In all reactions, we used 3.75–5 pmol of *Pf*Ago and 20–50 ng of the register. gDNAs were either phosphorylated using T4 Polynucleotide Kinase (NEB) in lab or phosphorylated guides were purchased from IDT. For each nicking reaction, a (2–10):1 ratio of guides to enzymes was formed. All guides were used in equimolar mixtures.

**Cas9 nickase experiments**. The Cas9 D10A nickase was purchased from IDT (Alt-R® S.p. Cas9 D10A Nickase); crRNAs were designed via IDT's Custom Alt-R® CRISPR-Cas9 guide RNA design tool. Both crRNAs and tracrRNAs were purchased from IDT and hybridized based on the manufacturer's protocol. The 10× Cas9 reaction buffer included: 200 mM HEPES, 1 M NaCl, 50 mM $MgCl_2$, 1 mM EDTA, pH 6.5. All Cas9n nicking reactions were set-up based on the manufacturer's protocol and performed at 37 °C for 60 min.

**Protocol verification via gel electrophoresis**. ssDNA gel analysis was performed using a 2% agarose gel. Nicked dsDNA samples were first denatured at high temperature (99 °C) for 10 min, and immediately cooled to 4 °C. The ssDNA products were then run on a pre-made 2% agarose Ex-Gel (Thermo Fisher Scientific).

**Sample preparation for MiSeq sequencing**. All nicked PCR products (obtained either via *Pf*Ago or Cas9n reactions) were purified using the Qiaquick PCR purification kit (QIAGEN) and eluted in $ddH_2O$. The dsDNA registers were denatured at 99 °C for 10 min, and immediately cooled down to 4 °C. The ssDNA samples were first quantified via the Qubit 3.0 fluorometer. Next, the Accel-NGS® 1S plus DNA library kit (Swift Biosciences) was used for library preparation following the manufacturer's recommended protocol. Prepared libraries were quantitated with Qubit, and then run on a DNA Fragment Analyzer (Agilent, CA) to determine fragment sizes, pooled in equimolar concentration. The pool was further quantitated by qPCR. All steps were performed for each sample separately and no nicked DNA samples were mixed.

**MiSeq sequencing**. The pooled libraries were loaded on a MiSeq device and sequenced for 250 cycles from each end of the library fragments with a Nano V2 500 cycles kit (Illumina). The raw fastq files were generated and demultiplexed with the bcl2fastq v2.20 Conversion Software (Illumina).

**Reference alignment**. Data was processed using a Nextflow-based workflow[26], implemented as follows. Sequence data was trimmed using Trimmomatic v.0.36[27] in paired-end mode using the options ILLUMINACLIP: adapters/TruSeq3-PE-2. fa:2:15:10 LEADING:20 TRAILING:20 SLIDINGWINDOW:4:15 MINLEN:20. Reads were aligned to the reference sequence using bwa v 0.7.10[28] with the command bwa mem -t 12 <REFERENCE><R1><R2>. Alignments were sorted and processed using samtools v1.6[29]. Insert size statistics were collected using Picard v.2.10.1[30]. Aligned files (BAMs) were then split based on expected fragment size using sambamba[31] with the option sambamba view -t 4 -f bam -h -F (template_length ≥ [LOWER] and template_length ≤ [UPPER]) or (template_length ≥ −[UPPER] and template_length ≤ −[LOWER]), with the upper and lower bound settings in brackets originally set to allow for one additional base greater and lesser than the expected size. Read coverage files were then generated using bedtools[32] and bedGraphToBigWig[33]. Alignment and coverage information was visualized in IGV v2.3.10[34]. All the scripts used for data analysis are available from the corresponding authors upon request.

**Nanopore simulations**. To obtain the trajectories of the nicked molecule translocating through the nanopore, all-atom Molecular Dynamics simulations were performed using NAMD[35]. For these simulations, the DNA structure (30 nucleotides around the 5th nicking site in the register) was obtained from the 3D-DART webserver[36] and described using the CHARMM27 force field[37]. Appropriate backbone molecules were manually removed to create the nicks in the desired locations of the strand. In order to obtain a stronger nanopore current signal from the DNA backbone, the $PO_3$ groups located at the nicked position may be removed by treatment of the nicked dsDNA with a phosphatase enzyme such as BAP (bone alkaline phosphatase).

The DNA molecule was placed just above the nanopore of a molybdenum disulfide ($MoS_2$) membrane to ensure a successful translocation process. The nanopore membrane and the biomolecule were then solvated in a water box with ions ($K^+$ and $Cl^-$) placed randomly to reach a neutrally charged system of concentration 1 M. Van der Waals energies were calculated using a 12 Å cutoff. Each system was minimized for 5000 steps and further equilibrated for 2 ps in an NPT ensemble, where the system was maintained at 1 atm pressure by a Langevin Piston[38] and at constant 300 K temperature using a Langevin thermostat. After equilibration, an external electric field was applied to the system in vertical direction to drive the nicked DNA through the nanopores.

A trajectory file of molecules driven through the nanopore by the applied electric field obtained from the MD simulations was used to calculate the ionic current via Eq. (1)[39], where $q_i$ and $z_i$ denote the charge and $z$-coordinate of ion i, respectively; $V$ denotes the voltage bias (1 V) and $L$ the length of the water box along the $z$-direction, while $N$ represents the number of ions and $\Delta t$ the interval between the trajectory frames:

$$I(t) = \frac{1}{\Delta t\, L_z} \sum_{i=1}^{N} q_i(z_i(t+\Delta t) - z_i(t)),$$

For each frame of the trajectory, the electrostatic potential is calculated using the following non-linear Poisson Boltzmann formula

$$\nabla \cdot [\varepsilon(r)\nabla\varphi(r)] = -e[C_{K^+}(r) - C_{Cl^-}(r)] - \rho_{DNA}(r),$$

where $\rho_{DNA}$ denotes the charge density of DNA, $\varepsilon(r)$ the local permittivity, and where $C_{K^+}(r)$ and $C_{Cl^-}(r)$ equal the local electrolyte concentrations of $K^+$ and $Cl^-$ and obey the Poisson–Boltzmann statistics. The detailed description of the method used is outlined elsewhere[40]. The calculated electrostatic potential is used to obtain the transverse sheet conductance in $MoS_2$ quantum point contact nanopore membranes. The electronic transport is formulated as a self-consistent model based on the semi-classical thermionic Poisson–Boltzmann technique using a two-valley model within the effective mass approximation. The calculated conductance at a given energy mode is described according to

$$G_{n_{1,2}} = \frac{2e^2}{h} \frac{1}{1 + exp\left(\frac{E^K_{n_{1,2}} - E^L_F}{k_{BT}}\right)} + \frac{2e^2}{h} \frac{1}{1 + exp\left(\frac{E^Q_{n_{1,2}} - E^L_F}{k_B T}\right)}$$

where $E^L_F$ denotes the quasi-Fermi level and is set depending on the carrier concentration (chosen to be $10^{12}$ cm$^{-2}$); in addition, $n_{1,2}$ represents the energy modes of the two conductance channels while $E^K_{n_{1,2}}$ and $E^Q_{n_{1,2}}$ stand for the energy modes at these two channels caused by the effective masses $K$ and $Q$, respectively. A detailed discussion of the thermionic current model is described elsewhere[41].

The $MoS_2$ membrane was modeled using VMD[42] from a basic unit cell. The Lennard–Jones parameters for $MoS_2$ are $\sigma_{Mo-Mo} = 4.4$ Å, $\varepsilon_{Mo-Mo} = 0.0135$ kcal/mol and $\sigma_{S-S} = 3.13$ Å, $\varepsilon_{S-S} = 0.3196$ kcal/mol, taken from Stewart et al.[43]. All $MoS_2$ atoms were fixed to their initial positions. Pore of diameter 2.6 nm is drilled by removing the atoms whose coordinates satisfy the condition $x^2 + y^2 \leq r^2$, where $r$ is the radius of the pore. Nick in each dsDNA strand were created by manually removing the phosphodiester bond and phosphate atoms in the backbone[44].

As a final remark, we observe that the simulations in Supplementary Figs. 12–14 indicating strong negative correlations of the global minimum and maximum of the sheet and ion current may be interpreted as follow: When nicked DNA translocates through the pore, the oscillations of the nicked backbone allow more ions to pass through the pore, leading to a steep increase (maximum) in the ion current. At the same time, the removed $PO_3$ group was saturated by an H atom which leads to a decrease in the sheet current to its global minimum.

**Reporting summary**. Further information on research design is available in the Nature Research Reporting Summary linked to this article.

## Data availability

The source data used to create Figs. 2–4 and Supplementary Figs. 1–17 is provided as part of the Source Data file. All sequencing data is available at SRA under number BioProject ID PRJNA612868. All other data are available from the authors upon reasonable request.

The data is available under the BioProject ID PRJNA612868, SubmissionID: SUB7154367, BioProject ID: PRJNA612868, Locus tag prefix: HA499 at http://www.ncbi.nlm.nih.gov/bioproject/612868.

The SRA accession number is PRJNA612868 with the release date: 2020-03-18. The SRA records is accessible at https://www.ncbi.nlm.nih.gov/sra/PRJNA612868.

## Code availability

All the codes used in this work are available from the authors upon reasonable request.

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

## Acknowledgements

This work was funded by the DARPA W911NF-18-2-0032 Molecular Informatics program. We thank SMH Tabatabaei Yazdi, and Jianhao Peng for many discussions and help. The authors also gratefully acknowledge supercomputing resources offered by Extreme Science and Engineering Discovery Environment (XSEDE) grant: TG-MCB170052 and Blue Waters grant: baxi.

## Author contributions

O.M. and H.Z. developed the nicking-based data recording platform. S.K.T. performed the nicking and toehold creation experiments and all other writing experiments. D.S., B.W., and O.M. designed the bitwise random-access system. D.S. and B.W. performed the bitwise access experiments. J.P.L. and N.A. implemented the nanopore simulations. A.G.H., D.S., O.M., and SK.T. designed the readout system, while A.G.H. performed all the MiSeq sequencing experiments. C.F. performed the MiSeq read analysis and reference alignment, while B.E. performed initial *Pf*Ago nicking activity verifications and helped with nicking experiment designs and protein purifications.

## Competing interests

University of Illinois at Urbana-Champaign has filed a (pending) nonprovisional patent on behalf of O.M., H.Z., A.G.H., and SK.T., with application number: 16/136,066. The patent is on nick-based data storage in native nucleic acids, and it includes part of the work pertaining to positional coding, sense and antisense nicking, comparative studies of nicking enzymes, orthogonal register selection and reading via NGS technologies. Other authors declare no competing interest
