## [Peer Review File · Nature Communications]

Reviewers' Comments:

Reviewer #1:

Remarks to the Author:

Tabatabaei et al. report a novel, alternative strategy to encode, store and retrieve information in DNA that can have several advantages over other current techniques also using DNA. This "paradigm", which holds the promise to be cheaper, more flexible and easier to scale up, is using E. coli native DNA that is modified by nicks at controlled positions of the DNA backbone produced by using the PflAgo restriction enzyme. The idea is appealing and the authors managed with the reported proof of principle to convince that it can be practically implemented for real life applications. However, before even going on the technical aspects, I found the manuscript in its current form to be confusing and not well structured. First, the manuscript is too concise and most of the real data are in supplementary information (SI): in the introduction one can find already part of the methodology and in the results only one single short paragraph is describing the proof of principle with reference to SI; then the authors jump to reading using solid-state nanopores using toeholds vs simple nicks and the problem of in-memory computation and bitwise random access that were not previously introduced; they end with a very short discussion that again makes references to a lot of material in SI and only poorly explains all the limitations and advantages of their paradigm. As a result, the reading is quite difficult, especially for a general reader as the target of Nat Comms, and the message does not pass clearly, which is a pity given the potential of the present work. I also had the feeling that the authors wanted to pack too much material in a single paper: for instance, while the first part on writing is very well executed and solid, I found the second part more speculative, as for instance nick reading by nanopores is based on simulations and not directly backed-up by nanopore experiments. The authors might want to consider to focus only on one aspect or develop better all the different topics they are discussing. The title for instance seems already to emphasize more the writing aspects.

Other more technical points relative to simulations that should be addressed:

- What is the meaning of "simulations based on quantum transport calculations"? MDs performed in this work look classical simulations based on empirical force fields.
- It is not very clear how the model systems are built for MD simulations. How is the MoS₂ membrane modeled? what is the diameter of the pore? which exact molecules of the backbone were removed to produce the nick? how were they capped with respect to intact dsDNA? All this information needs to be included and discussed to ensure reproducibility of the results.
- MD equilibration and production runs seem also to be extremely short, with very fast translocation time for DNA. Sampling in these simulations is poor: only one voltage condition is tried; only 3 different DNA sequences are used. While it is appealing to infer that the negative correlation between the global minimum of the transverse sheet current and the maximum of the ion current is a proxy for nick reading, the results of these simulations are not statistically significant to confidently state it. Already simulations reported in Figures S13 and S14 do not have a clear readout as S12. Given the short timeframe of these MDs, many more events should be sampled by MD and analyzed.

Minor things:

- 1210...0 in caption of figure 2 should be 121...0
- file sizes in table S3 are inverted

Reviewer #2:

Remarks to the Author:

The manuscript by Tabatabaei et al. describes the usage of nicking enzymes to store information

in pre-fabricated (natural) DNA sequences. While the idea is attractive, and should eventually be published in a multidisciplinary journal, the manuscript is quite problematic. It does contain some interesting experimental results (especially the performance of the nicking enzymes), but a considerable part of the manuscript is a mixture of ideas, and the conclusions are not supported by the experimental (or theoretical) data shown. In addition, the experiments are not described so that they can be reproduced, and parts of the cost calculation seem to be erroneous.

It would be useful if the authors would focus a larger part of the manuscript on the actual DNA writing process, the activity and performance of the enzymes, instead of speculation on future readout technologies, and the single-bit random access, which does not seem to be scalable. Secondly, a comparison with state of the art DNA storage is useful, but the authors should rather compare actual data and numbers, than speculate towards future developments. If the authors use speculative and include future ideas into their conclusions, this should be clearly stated so that readers can differentiate conclusions based on scientific data and potential future outcomes.

Details:

Single bit toehold readout:

The toehold readout works for single DNA sequences in a pool, and might also work on the usage of orthogonal DNA sequences combined in a pool. However, it does not seem scalable, as it is not compatible with the ideas of combinatorial register mixing. Also, detection of toehold replacement requires large quantities of nicked DNA, and nicked DNA cannot be amplified (keeping the nicks). As a result, single-bit toehold readout via fluorescence as shown does not seem possible at the high storage densities calculated in the table (i.e. when only very low copy numbers of nicked DNA are present). As a result, it does not seem very useful. Can the authors give application examples, where this would be useful in the context of data storage at high storage densities? Otherwise, this should be moved to the supp. Info.

Several questions on the proof of concept experiments, as the wording used in the description is not very clear:

- Did the authors actually read and decode the LGA and LMI recordings, other than the reported random reads? If yes, please give the experimental procedures, including data representation on the DNA strands, combinatorial register mixing and multiplexing. Can the 1024 strands be read together in a single read?
- Data stored: is 14 kB the size of the compressed or uncompressed files? Please only give data of size of actually stored data (in this case compressed data). It is not quite clear how 1024x 10 bit strings can be used to store 14 kB of Shannon data.
- What is the actual consumable cost of storing this dataset?

The table in the manuscript and some of the values reported therein are speculative, and do not seem to be computed with scientific rigor. Some details:

Data density:

In the current thinking of using DNA as a data storage tool, the DNA is synthesized, and then diluted. 10 – 1000 fold physical DNA coverage has been shown to be useful. Such a DNA pool can be quite large (at least 200 MB in one pool have been shown experimentally, proof of concept of random access in significantly larger pools has been also shown (e.g. Tomek et al. ACS Synth. Biol 2019)). With the novel approach shown, it seems that the DNA generated cannot be combined at will, and certainly not at the scale of experiments performed for synthetic DNA, but has to be kept separate in microwells. This has a massive impact on the data density, as the carrier of the DNA (e.g. the microwell shown in Fig 1) should also be taken into account when calculating the data

density.

Nicked DNA cannot be directly amplified via PCR. Can the authors give information on how much DNA is required in the preparation of the sequencing runs. How much physical coverage is required for readout? Currently the experimental part fully lacks details on concentrations. From the broad range of coverages reported for the individual strands (Fig 2c), it seems that many physical copies of every strand are required for the reading to work.

Pricing:

It is not correct that the costs of the enzyme and E.coli derived DNA can be excluded from the calculations. Especially PCR and consecutive DNA purification are relatively expensive operations.

Per PCR well, the authors can perform > 100 reactions. The cost of a PCR well is at least 0.2 USD for the master mix alone, and the purification of the DNA with the used kits is > 1 USD per well. Even assuming future performance increases, the DNA adds 0.01 USD in cost per reaction. Can the authors give the actual costs of the PCR products used in the study?

With one reaction the authors can currently store 10 bits of data, yielding the cost per bit at least 0.001 USD. This does not include pipetting tips or any similar consumable used, and is still many orders of magnitude larger than the number stated in the manuscript.

Equation on page 44:

For 24 USD the authors purchased 30 nmols of 10 different guide DNAs each, yielding 300 nmol of guide DNA. To write 10 bit, which is equal to 10^{24} nicking sites, the authors require 7.5 pmol total of guide DNA per reaction.

Consequently, the calculation on page 44 does not seem to be correct. The authors can perform $300\text{nmol}/7.5\text{pmol}$ reactions, which is 40000. Each reaction gives 10 bit of information, and consequently 400000 bits of information can be generated by spending 24 USD on guide DNA. This results in a cost per bit of 0.00006 USD/bit, which is 100 times higher than the number given in the table. (even without considering costs of e.coli DNA etc.)

Also, what is the cost of 5'-phosphorylation? Shouldn't this cost be included in the cost of the guide DNA?

DNA reading:

Can the authors also give some data on the cost of reading (compared to traditional DNA data storage?) It seems that the large difference in sequence coverage between the individual sequences observed in the data will require considerable sequencing depth. E.g. the data reported in Fig 2c requires > 10000 sequence reads so that the sequence segment with the smallest coverage is also detected. Side note: In this figure, how is a sequencing depth of 1.18 computed?

Do the authors have an explanation for the large differences in the sequence coverages observed?

It is obvious that solid state nanopore sequencing may result in an improved reading of nicks, but if this is cost effective and scalable is still speculative. The authors might want to keep the theoretical work as part of the manuscript (preferably supp. Info), but state clearly that the combination of solid state nanopore sequencing and the proposed technology lie in the future.

Conclusions:

The following sentence is highly problematic:

„It also allows for cost-efficient scaling as a. long registers and mixtures of orthogonal registers may be nicked simultaneously; b. most uncompressed data files do not contain all possible 10-mers or compositions of orthogonal k-mers; c. genomic DNA and PfAgo, as the writing tool, are readily available...“

a) It should read „With future developments, it may allow for cost-efficient scaling...“

b) Why would the authors want to store uncompressed data? The experiments were performed on compressed data.

c) As discussed above, the cost of genomic DNA at sufficient quality is an important factor and is not "readily available"

The last sentence of the manuscript is not based on any data or discussion shown, and should be written to indicate the speculative nature of the content. (e.g. „This storage system might...“)

Response to Reviewer 1:

“Tabatabaei et al. report a novel, alternative strategy to encode, store and retrieve information in DNA that can have several advantages over other current techniques also using DNA. This “paradigm”, which holds the promise to be cheaper, more flexible and easier to scale up, is using E. coli native DNA that is modified by nicks at controlled positions of the DNA backbone produced by using the PflAgo restriction enzyme. The idea is appealing and the authors managed with the reported proof of principle to convince that it can be practically implemented for real life applications. However, before even going on the technical aspects, I found the manuscript in its current form to be confusing and not well structured. First, the manuscript is too concise and most of the real data are in supplementary information (SI): in the introduction one can find already part of the methodology and in the results only one single short paragraph is describing the proof of principle with reference to SI; then the authors jump to reading using solid-state nanopores using toeholds vs simple nicks and the problem of in-memory computation and bitwise random access that were not previously introduced; they end with a very short discussion that again makes references to a lot of material in SI and only poorly explains all the limitations and advantages of their paradigm. As a result, the reading is quite difficult, especially for a general reader as the target of Nat Comms, and the message does not pass clearly, which is a pity given the potential of the present work. I also had the feeling that the authors wanted to pack too much material in a single paper: for instance, while the first part on writing is very well executed and solid, I found the second part more speculative, as for instance nick reading by nanopores is based on simulations and not directly backed-up by nanopore experiments. The authors might want to consider to focus only on one aspect or develop better all the different topics they are discussing. The title for instance seems already to emphasize more the writing aspects.”

We thank the reviewer for an in-depth reading of the text and for her/his comments. We agree that the organization of the manuscript can be improved and that too few details were provided in the main text about the various experiments performed. In the revised manuscript, we have addressed these issues as follows.

First, we included a more in-depth description of the system in the Introduction, Results and Discussion Sections.

The introduction now includes a detailed description of each step in the recording and reading process and a simple summary of the operational properties of the system.

In the Results Section, we introduced several subsections to better organize the content. In Subsection “The Write Architecture” we outlined the steps in the recording process, including: Choosing the right writing tool; DNA register and nicking sites selection; Positional encoding; Data recording and data organization; Native DNA extraction (which does not need significant explanation); Selection of the registers and nicking sites; and Design of guides and enzymes. We also briefly describe the combinatorial mixing approach for registers and sense-antisense nicking processes as means for improving the recording density. Although we agree with the reviewer that the

writing architecture is the most original component of the system, we still believe that a description of a practical storage device has to include the readout architecture. Hence, in the Subsection “The Read architecture” we explain our new readout procedure that relies on sequencing fragments between nicks and aligning the reads with the readily available native reference or references (as multiple registers can be mixed and sequenced together). Furthermore, we describe a potentially useful portable, cost-effective and non-destructive readout system centered around solid-state nanopores. Although at this point, we only have MD simulations supporting our claims, we would like to point out that many lines of research on solid state nanopores are supported only by simulations. This is due to the fact that some critical components in the sequencers, such as molecular motors, are still missing. Nevertheless, there is a strong belief that these issues with solid state nanopores will be resolved in the near future. As an additional piece of evidence, we cited our companion paper reporting successful experimental detection of single-stranded toehold regions in double stranded DNA using solid state nanopores. The work is listed as reference [22] in the text and it appeared in Nature Communications this year.

We have also added another subsection named “Additional Functionalities” in which we explain other features of our system, including sense-antisense recording, multi-register recording, combinatorial mixing and bitwise random access.

Answers to specific questions regarding our MD simulations:

“What is the meaning of “simulations based on quantum transport calculations”? MDs performed in this work look classical simulations based on empirical force fields.”

Answer: We thank the reviewer for correctly pointing out this issue. This is an inadvertent typo in the previous version of the manuscript. The text should have read as follows: “We performed Molecular Dynamics (MD) simulations along with quantum transport calculations to obtain the ionic and transverse current signals.” instead of “We performed Molecular Dynamics (MD) simulations based on quantum transport calculations”.

“It is not very clear how the model systems are built for MD simulations. How is the MoS₂ membrane modeled? what is the diameter of the pore? which exact molecules of the backbone were removed to produce the nick? how were they capped with respect to intact dsDNA? All this information needs to be included and discussed to ensure reproducibility of the results.”

Answer: The reviewer is once again justified in asking for a more detailed description of the platform. We now included the following paragraph in the Methods Section:

“The MoS₂ membrane was modeled using VMD [43] from a basic unit cell. The Lennard–Jones parameters for MoS₂ are set to $\sigma_{\text{Mo-Mo}} = 4.4 \text{ \AA}$, $\epsilon_{\text{Mo-Mo}} = 0.0135 \text{ kcal/mol}$ and $\sigma_{\text{S-S}} = 3.13 \text{ \AA}$, $\epsilon_{\text{S-S}} = 0.3196 \text{ kcal/mol}$, taken from [44]. All MoS₂ atoms were fixed to

their initial positions. A pore of diameter 2.6 nm was drilled by removing the atoms whose coordinates satisfy the condition $x^2 + y^2 \leq r^2$, where r is the radius of the pore. The nicks in each dsDNA strand were created by manually removing the phosphodiester bond and, in addition, the phosphate atoms in the backbone [45]”

For more details, please also refer to the Section Nanopore Simulations in the Methods Section.

“MD equilibration and production runs seem also to be extremely short, with very fast translocation time for DNA. Sampling in these simulations is poor: only one voltage condition is tried; only 3 different DNA sequences are used. While it is appealing to infer that the negative correlation between the global minimum of the transverse sheet current and the maximum of the ion current is a proxy for nick reading, the results of these simulations are not statistically significant to confidently state it. Already simulations reported in Figures S13 and S14 do not have a clear readout as S12. Given the short timeframe of these MDs, many more events should be sampled by MD and analyzed.”

Answer: The main issue with performing more simulations is the fact that the time needed to run more instances of the strings is excessively high, and we had time-restricted access to the Blue Waters supercomputer. Hence, while it is true that we have not simulated more than three events, the purpose of the work was to provide a proof-of-principle for directly detecting nicks using MoS₂ solid-state nanopore devices. In order to reduce the computational time of the MD simulations, DNA strands of 30 nucleotides with only one nicked site in the middle were considered. Given that the simulations were performed at 1 V, the total translocation time was very fast (few ns). Though the current signals shown in Figure S13 and S14 do not have the same readout as Figure S12, the negative correlation still exists between the signals when the nick resides in the pore. In order to clearly identify the presence of a nick, we provided additional analysis by calculating the Pearson’s correlation between the normalized signals to indicate the time frame when the nick is residing within the pore. The correlations are depicted in Figure S12c.

We would also like to thank the reviewer for pointing out the typos in the manuscript which now have been fixed.

Response to Reviewer 2:

“The manuscript by Tabatabaei et al. describes the usage of nicking enzymes to store information in pre-fabricated (natural) DNA sequences. While the idea is attractive, and should eventually be published in a multidisciplinary journal, the manuscript is quite problematic. It does contain some interesting experimental results (especially the performance of the nicking enzymes), but a considerable part of the manuscript is a mixture of ideas, and the conclusions are not supported by the experimental (or theoretical) data shown. In addition, the experiments are not described so that they can be reproduced, and parts of the cost calculation seem to be erroneous.

It would be useful if the authors would focus a larger part of the manuscript on the actual DNA writing process, the activity and performance of the enzymes, instead of speculation on future readout technologies, and the single-bit random access, which does not seem to be scalable. Secondly, a comparison with state of the art DNA storage is useful, but the authors should rather compare actual data and numbers, than speculate towards future developments. If the authors use speculative and include future ideas into their conclusions, this should be clearly stated so that readers can differentiate conclusions based on scientific data and potential future outcomes.”

We agree with the reviewer that the paper contains a mixture of ideas, but all of them are necessary to describe all possible diverse functionalities of a modern storage system. The reviewer points out that the writing platform is interesting, but equally important in practice is the readout system. The readout system is part of the user platform and needs to be portable, delay- and cost-efficient. Furthermore, current storage systems offer random access at different levels, while prior DNA storage implementations have only succeeded in ensuring access to blocks of data (e.g., oligos).

In particular, we are not sure which parts of our results are speculative, or not supported theoretically. The theoretical results pertain to group testing for mixing of registers and have a very firm theoretical grounding, as described in the Supplement. All experiments can be reproduced, and a very detailed explanation of the various experimental steps is available in the Supplement. With respect to the cost calculations, we tried our best to avoid speculating about future pricings; still, we felt the need to mention why we believe that the systems, when properly scaled up, would be advantageous compared to synthetic DNA platforms for many storage applications. All we did is to list the exact amounts we paid for various reagents, toolkits, enzymes, etc. Admittedly, we did not include what we considered to be minor expenses, but we addressed this issue in the revised manuscript (see our detailed response below).

For the bitwise random-access scheme, the reviewer is correct that the scheme cannot be used for mixed nonorthogonal pools. Nevertheless, it can be easily used in pools that contain one or multiple orthogonal registers as previously described in the Supplement and now moved to the main text.

We have a large degree of freedom in choosing orthogonal registers, which are nothing more than ordered native DNA strands with large edit distances between themselves. Orthogonal registers enable scaling the system, and in our setting, we mixed five orthogonal registers to demonstrate this fact. All chosen registers have a small number of toehold sites (between 5 and 10) and the sequence content ensures specific access to the toehold locations. Hence, the combination of the reporter sequence and the fluorophore provides an easy way to scale the system.

In more detail, since the positions of the nicks are predetermined and since orthogonal registers are mixed, one only needs to have one reporter sequence available for each bit location. For the running example register, we need only ten different fluorophores for bitwise random access to all ten positions simultaneously. If access is sequential, only one fluorophore is needed. This is to be contrasted with the first random access scheme developed by our group proposed and reported in [4] – there, one can only access individual oligos through a time-consuming process of PCR and sequencing.

This being said, we added more information into the text to clarify the relevance of each component. Based on the reviewer's comments, in the revised manuscript we added a more in-depth discussions of the enzymatic reactions, the encoding scheme, the selection process for the registers and nicking sites, as well as pertinent details regarding the readout and random-access system. Also, we changed the order of our exposition, starting with an overview of DNA-based data storage methods, followed by a detailed description of our system. This should make the introduction easier to read. The results section now also includes “multi-register recording” and “sense-antisense nicking” experiments along with separate figures plotting the corresponding results. The two aforementioned approaches allow for increasing the storage density per register and per DNA spot. To make the content better organized, we introduced subsections.

Our additional explanations mostly focus on the writing architecture. However, as already pointed out, no storage system is complete without a solid and efficient readout architecture. This is why we choose to keep the sections discussing the sequencing procedures: high-throughput sequencing experiments followed by reference-based alignment and simulations pertaining to solid-state nanopore readout platforms.

Answers to specific questions:

“Single-bit toehold readout: The toehold readout works for single DNA sequences in a pool and might also work on the usage of orthogonal DNA sequences combined in a pool. However, it does not seem scalable, as it is not compatible with the ideas of combinatorial register mixing. Also, detection of toehold replacement requires large quantities of nicked DNA, and nicked DNA cannot be amplified (keeping the nicks). As a result, single-bit toehold readout via fluorescence as shown does not seem possible at the high storage densities calculated in the table (i.e. when only very low copy numbers of nicked DNA are present). As a result, it does not seem very useful. Can the authors give application examples, where this would be useful in the context of data storage at high storage densities? Otherwise, this should be moved to the Supp. Info.”

Answer: As already pointed out, in mixtures involving the same registers with different nicking patterns, one cannot use toeholds directly to enable random access. But for mixtures of orthogonal registers, random access is possible and straightforward to achieve. It is also worth pointing out that in the mixture case, the information is not really stored in individual nicks/toeholds but in a combinatorial manner.

In particular, our experiments for the first time show that in addition to nicks, one can also create toeholds (single-stranded gaps in double-stranded DNA) through parallel enzymatic nicking. Nicks and toeholds enable strand displacement reactions which are required for many molecular computing paradigms (see references [24-26]). Hence, one role of the toeholds is to enable computing, and this line of work is discussed in our companion paper [15]. In addition, toeholds allow for bitwise random access through the use of site-specific reporter DNA sequences with quenchers and fluorophores. If a toehold represents one bit, it is possible to tell if that bit is zero or one through the use of fluorimeters. Furthermore, the fluorescence intensity carries information about the concentration of registers that have a specific bit value, which is helpful in pooling and search applications. And although the system at first glance may not appear scalable, it is important to notice that we are dealing with registers that have a small number of distinct nicking sites (≤ 10) and that we hence only need to have ten different reporters. These reporters may have the same fluorophore if random access is sequential or may require different fluorophores if one wants to access multiple locations in parallel. In this case, it still suffices to have only ten different fluorophores, and these can be easily obtained from IDT (see IDT.com, Fluorophores Modifications). Note that the same fluorophore may be used for detecting toeholds in different orthogonal registers since the fluorophores are attached to reporter strings specific to the register content.

Another application of toeholds we did not discuss in our paper to avoid clutter is for storing categorical data: a nick at a certain position may indicate that the data stored has some property (e.g., data pertaining to male/female subjects etc). In that case, searching for a data sample with a given categorical property may be accomplished in the same way as random access. Most importantly, the access procedure is **completely non-destructive** as one does not need to sequence the data. This is of great importance since all current DNA storage systems only allow for several rounds of reading due to the destructive nature of the process.

“Several questions on the proof of concept experiments, as the wording used in the description is not very clear: Did the authors actually read and decode the LGA and LMI recordings, other than the reported random reads? If yes, please give the experimental procedures, including data representation on the DNA strands, combinatorial register mixing and multiplexing. Can the 1024 strands be read together in a single read?”

Before actually encoding LGA and LMI we performed four tests on the running example register, including one nick, three, five and ten nicks. These tests were conducted to compare the accuracy of the nicks produced by *PfAgo* and Cas9. The sample quality was assessed through gel analysis in order to avoid more expensive sequencing procedures,

and for 2 samples additional sequencing tests were performed. These results are documented in the Supplement and we prefer to keep them in the same part of the text. Upon seeing positive gel results, we proceeded to create the complete library of 1024 nicking patterns for the example LGA register with *PfAgo*. Out of these 1024 patterns, we randomly selected 8 patterns used in the encoding of LMI and LGA. Those 8 registers were sequenced, while others were not so as to avoid large expenses. These are all documented in the Supplement. Furthermore, we ran three additional sequencing experiments. One experiment included preparing 4 mixtures of five different orthogonal registers containing a total of 126 nicking positions (bits). The results for one of the mixtures are included in the main text, in a separate figure, while the remaining results are included in the Supplement. The second sequencing experiment served as a proof of concept for sense-antisense nicking, where we nicked the sense strand in 6 positions and the antisense in 4 positions. Only one such sample was sequenced, and the results of this test have now been moved to the main text. The final set of tests involved mixing the same register sequences with different contents using a group testing schemes; 8 samples were sequenced for these experiments, as documented in the Supplement.

In addition, we performed one sequencing experiment to determine if parallel nicking can lead to the creation of the toehold.

In summary, we sequenced 24 samples and none of them contained reconstruction errors. We hope that the reviewer agrees that sequencing all 11520 registers individually would have been prohibitively expensive as we already ran a large number of sequencing experiments to test our preliminary implementations. It is important to point out that one sample included all ten nicks, which was crucial to establish that parallel nicking at all ten sites simultaneously is possible to do in a highly accurate manner.

Furthermore, arbitrary combinations of subset of the 11520 register nicking patterns cannot be read together using MiSeq as one would not be able to properly align the reads to one reference. We explained under which conditions the registers with different patterns may be sequenced together (via the introduction of group testing schemes described in the supplement) and performed 8 experiments (described above) to confirm that the scheme works. Nevertheless, if sequencing can be performed via nanopores, which we hope to be able to do in the near future, arbitrary register mixing will be possible.

“Data stored: is 14 kB the size of the compressed or uncompressed files? Please only give data of size of actually stored data (in this case compressed data). It is not quite clear how 1024x 10 bit strings can be used to store 14KBs of data.”

As mentioned in the main text, the stored data was compressed – the detailed information was provided in the Supplement, Table S.3. It is important to point out that we did not store the 14KBs in the 1024x wells containing the nicked registers. We actually organized the registers into rectangular grids of microwells/microspots to enable additional random access to groups of ten bits. This is what may have created confusion – the figure in the introduction clearly explains how the nicked registers are placed. Clearly, the spots can contain mixtures of orthogonal registers and properly selected mixtures of the same register with different nicking patterns. To avoid using separate

spots for the register one can use special, new coding methods which we will discuss in a companion paper.

Note that encoding compressed data in synthetic DNA is challenging since errors in compressed formats tend to exhibit catastrophic error propagation. Hence, our error-free nicking scheme is exceptionally well-suited for compressed data representation. Other methods cannot avoid adding significant coding redundancy to mitigate the effects of error propagation.

“What is the actual consumable cost of storing this dataset? “

To store the reported datasets (14.4 KB) we actually took a nonstandard approach and created the whole library of 1024 different nicked registers; from this library, we selected registers to be mixed and placed on the microplates/spots as dictated by the datasets. Given that the library is of a size significantly larger than what was needed to encode the image and text, we added calculations that explain how large of a file could have been stored with the given library. These calculations are now updated and reported in Section B.10 of the Supplement. Based on the reviewer’s request, we added the cost of microplates to the overall expense list: For the files in question, one needs 30*384-well plates, with a total cost of $\sim 30 * \$6.5 = \195 , or 8*1536-well plates, with a total cost of $\sim 8 * \$23 = \184 . Of course, these costs can be reduced when bulk purchases are made or by using specialized microspots for storing the registers akin to those used in DNA microarrays. Furthermore, as already pointed out, this is only a “transitional solution” as we do not expect to store the mixed content in bulky microplates (our ongoing research is focusing on micro and nano-level electrically controlled DNA capture-hold-release platforms; the results of this work will be reported elsewhere). Additional costs include the charges for guides ($10 * \$2.4 = \24 for a total of 300 nmols of unphosphorylated guides and $10 * \$18.4 = \184 for a total of 300 nmols of already phosphorylated guides, both purchased from IDT). For each reaction creating 10 bits, we used 7.5 pmols of guides. This amounts to a cost of \$51.5 for the guides. Again, one can purchase the DNA guides in bulk or perform in-house synthesis to reduce this cost. We still believe that DNA extraction and amplification as well as enzyme extraction can be performed with small cost if done in-house and at large scale. Further explanations regarding the cost calculations may be found in Section B.10 of the Supplement.

“The table in the manuscript and some of the values reported therein are speculative, and do not seem to be computed with scientific rigor. Some details: Data density: In the current thinking of using DNA as a data storage tool, the DNA is synthesized, and then diluted. 10 – 1000 fold physical DNA coverage has been shown to be useful. Such a DNA pool can be quite large (at least 200 MB in one pool have been shown experimentally, proof of concept of random access in significantly larger pools has been also shown (e.g. Tomek et al. ACS Synth. Biol 2019). With the novel approach shown, it seems that the DNA generated cannot be combined at will, and certainly not at the scale of experiments performed for synthetic DNA but has to be kept separate in microwells. This has a

massive impact on the data density, as the carrier of the DNA (e.g. the microwell shown in Fig 1) should also be taken into account when calculating the data density.”

We thank the reviewer for her/his comments, but we respectfully disagree that the table is speculative and “computed without scientific rigor.” Every entry in the table except for one entry describes well-documented system parameters discussed in great depth in the Supplement. The overall cost of the system may be a point of contention because it is very hard to take into account every minute expense made as part of the experimental testing. We actually made an effort to address all the steps in the writing and reading procedure and now have an updated, highly conservative new cost estimate. To not obscure the most important fact about our system – which is its unique ability to store data without resorting to sequential and time-consuming DNA synthesis by commercial vendors and make it an easy-to-implement in house product – we moved the cost analysis to the Supplement.

With regards to the microplates: The reviewer is correct that micro-plates are neither the optimal nor the cheapest way to organize the data, but as already pointed out, we see these as a transient solution. This work is a proof-of-concept for a completely new storage strategy and it is only to be expected that each component in the system will be improved in the future.

Also, it is important to notice that this team came up with the original idea of random access via PCR (Yazdi et al. *Sci. Reports*, 2015) and actually proposed an optimal information-theoretic strategy for address design. The issue is that even with optimal primer designs, PCR-based solutions will have severe limitations due to the need for large address spaces; to increase the number of useful addresses one needs to increase their length which effectively reduces the information content stored. Furthermore, PCR for random access is time and resource consuming and often produces erroneous results (after a few rounds of PCR on data encoded without using our specialized addressing procedure, some non-specific fragments of DNA begin to propagate through the amplification process; the same issue may be encountered with our implementation from 2015 if the address space becomes very large), while just sampling DNA from a well or microspot will not lead to the same problems. Ideally, these two methods should be combined, which is what we hope to accomplish in the future.

It is also important to note that all other known solutions have similar data organization and cost issues. An example includes (Kennedy et al. *Plos One*, June 2019), which reports storing data stored in metabolites in form of dots on MALDI plates. Some other solutions use microfluidic devices which can increase the data density in comparison to micro-plates, but such devices are expensive, and often hard-to-handle as they require special equipment for liquid transfer and visualization (Newman et al. *Nature Communications*, April 2019).

Therefore, we believe that our current “plate-based” prototype for positional random access is an acceptable solution. Currently, we can facilitate specific access to data both based on their well position and based on their content, through the use of toehold bitwise random access. Note that this organization also makes the process of reading less destructive than when storing the whole oligo pool in one well, since access is achieved via amplification of desired strings which effectively renders every other string as noise and makes it unusable for other reactions.

“Nicked DNA cannot be directly amplified via PCR. Can the authors give information on how much DNA is required in the preparation of the sequencing runs. How much physical coverage is required for readout? Currently the experimental part fully lacks details on concentrations. From the broad range of coverages reported for the individual strands (Fig 2c), it seems that many physical copies of every strand are required for the reading to work.”

We clearly agree that nicked DNA cannot be directly amplified via PCR, but the point to be made is that native DNA can be extracted in massive volumes. Given that our nicking enzymes are multiple turnover and can potentially make hundreds of nicks in parallel on the native substrates, one can move the burden of amplifying the substrates post-nicking to amplifying the registers pre-nicking. The readout process used in our high-throughput sequencing solution follows similar steps as that used in synthetic DNA-based storage methods. The difference lies in the fact that our DNA is double-stranded and that it has to be denatured and that the denaturation process produces nonuniform length fragments. More precisely, the nicked DNA samples are first converted to ssDNA and the ssDNA pool is used for dsDNA library preparation and subsequent sequencing. Those steps are exactly the same as the ones used in synthesis-based methods, in which synthetic oligos are amplified, converted to a library of dsDNAs and then sequenced. For our scheme, the library preparation kit of choice was Accel- NGS 1S Plus from Swift Biosciences, and it supports inputs as low as 10 pg, which is completely affordable at this scale. After library preparation (10 cycles of amplification), we usually obtained 50-100 nmols of DNA and diluted it in 600 ul of the appropriate buffer solution, which means a concentration of 4.2 nM. This is what is presented to the sequencers.

Regarding physical coverage, further explanation may be needed. In all our reported MiSeq sequencing results, we retrieved ~1 million reads. The sequenced pools were analyzed to obtain the frequency of each observed read (sequence) in a pool (i.e., through the use of insert size histograms). The frequency peaks are compared with the ssDNA fragment lengths that we expect to see after nicking, based on our choices of nicking sites. In all cases, for every fragment length we observed a sharp frequency peak in the insert size histogram. We interpret these peaks as indicative of the presence of nicks. For all the tests we ran, the least coverage that we observed was 2, and the latter case occurred for a 76 bp fragment (Figure 3c). Nevertheless, even this relatively small coverage can be resolved without errors given that the fragment lengths are co-dependent and that the fragments have to jointly “tile/cover” the known reference DNA

string. In simple terms, if one fragment has low coverage but leaves a hole in the aligned strings with respect to the reference, its presence is noted indirectly.

One additional observation is that the effective cost of NGS sequencing increases compared to the synthetic DNA approach given that we have to sequence a large symbol content in order to determine a small number of nicks. Effectively, we retrieve fewer bits for the same sequencing cost compared to synthetic platforms. This is why we suggested and computationally demonstrated the utility of nanopores and provided a reference to our companion paper that addresses the readout issue with nanopores through an experimental study. Finally, it is worth pointing that our scheme has a naturally built in robustness that is almost certainly guaranteed to produce error-free reconstruction in the presences of sever sequencing errors which is not the case for synthetic DNA platform. We hence do not have to add costly error-correction redundancy to resolve either missing oligos or sequencing errors.

“Pricing: It is not correct that the costs of the enzyme and E.coli derived DNA can be excluded from the calculations. Especially PCR and consecutive DNA purification are relatively expensive operations.”

“Per PCR well, the authors can perform > 100 reactions. The cost of a PCR well is at least 0.2 USD for the master mix alone, and the purification of the DNA with the used kits is > 1 USD per well. Even assuming future performance increases, the DNA adds 0.01 USD in cost per reaction. Can the authors give the actual costs of the PCR products used in the study?”

“With one reaction the authors can currently store 10 bits of data, yielding the cost per bit at least 0.001 USD. This does not include pipetting tips or any similar consumable used, and is still many orders of magnitude larger than the number stated in the manuscript.

Equation on page 44.”

The reviewer is correct that we neglected some minor expenses and we apologize for that issue – it is hard to account for consumables such as pipette tips. The cost is roughly 9 cents per tip, and we used 5 reagents and consequently five pipette tips per reaction. The reviewer should be once again reminded of the fact that the authors are academics who do not have access to bulk products. We still firmly believe that through collaborations with appropriate commercial vendors our costs will be as predicted or only slightly higher. More precise arguments are provided below.

1. It is true that we used commercially available kits and enzymes for DNA extraction to get our results as quickly as possible. But, as the reviewer probably knows, these toolkits are not necessary for extraction, they only made our experimental efforts less time consuming. For DNA extraction from different types of cells, including complex plant and mammalian cells, there are manual methods which can lead to a very high yield of DNA at a low-cost. See for example [<https://bio-protocol.org/bio101/e97#biaoti1286>].

2. For PCR amplification, we used the Q5 polymerase, which is a highly optimized enzyme, and therefore costlier than other enzymes that can be used for the same purpose. As an example, we could have used the Taq polymerase which can be purified cheaply in any lab with high yield. We did not use this polymerase as we wanted to obtain our results as quickly as possible to demonstrate the utility of our new storage scheme. Again, even if we use cheap enzymes that potentially introduce errors, we would not expect to see the errors in sequence content influence our nicked information as it is encoded in a different dimension. Studying every option and combination of enzymes and reagents etc. is clearly beyond the scope of the paper.

Of course, we also agree that at this scale of implementation, even minor consumables such as pipette tips can significantly contribute to the overall expenditure. However, these consumables are used in every biological experiment and these two reasons are why we did not include these expenses into our calculations.

“For 24 USD the authors purchased 30 nmols of 10 different guide DNAs each, yielding 300 nmol of guide DNA. To write 10 bit, which is equal to 1024 nicking sites, the authors require 7.5 pmol total of guide DNA per reaction.

Consequently, the calculation on page 44 does not seem to be correct. The authors can perform 300nmol/7.5pmol reactions, which is 40000. Each reaction gives 10 bit of information, and consequently 400000 bits of information can be generated by spending 24 USD on guide DNA. This results in a cost per bit of 0.00006 USD/bit, which is 100 times higher than the number given in the table. (even without considering costs of e.coli DNA etc.)”

As already pointed out in the Supplement, there is no highly accurate and direct way of comparing our platform with conventional synthesis-based systems. However, our intention was to give the readers a sense of how cheap this method of data storage could be once proper scaling and automation is possible. This being said, we indeed made an error in our calculation as we included a multiplicative factor of 1024 which was unnecessary since one reaction was pertaining to one register and not 1024 registers. The calculations should be done as follows: $(300 \text{ nmol}/7.5 \text{ pmol}) \times \text{bits per mixture (10)} = 400,000 \text{ bits}$ which is 100 times less than what was originally provided. This number is still >2 orders of magnitude lower than the commercial cost of a “DNA bit”.

“Also, what is the cost of 5'-phosphorylation? Shouldn't this cost be included in the cost of the guide DNA?”

To answer the reviewer's question about 5' phosphorylation, although we have used a commercially available enzyme (PNK), again we could have performed this reaction with in-house-extracted enzymes of considerably smaller cost. Another approach would be to

just order 5'-phosphorylated guides from IDT which cost ~7.6 times more (\$18.4 vs \$2.4). The latter solution will increase the overall cost.

Considering the concerns raised by the reviewer and the fact that the costs can only be predicted with limited accuracy for bulk purchases, we decide to move the cost calculations and comparisons to the SI and to clearly indicate that these are bulk purchase predictions and estimates. Also, given the fact that no other work on DNA-based data storage has reported similar cost calculations, there is no reliable source of information that would enable a highly accurate comparative study. We still believe that with or without the cost computations this work contains completely novel practical and theoretical implementations of DNA storage systems and ideas that will be of significant interest for the Nature Communications readership.

“Can the authors also give some data on the cost of reading (compared to traditional DNA data storage?) It seems that the large difference in sequence coverage between the individual sequences observed in the data will require considerable sequencing depth. E.g. the data reported in Fig 2c requires > 10000 sequence reads so that the sequence segment with the smallest coverage is also detected. Side note: In this figure, how is a sequencing depth of 1.18 computed? Do the authors have an explanation for the large differences in the sequence coverages observed? “

We do not believe that large sequencing depths are needed at all since our sequencing is not done de novo – we always have the reference at our disposal and this is what allows us to actually get away with very low coverage (unlike the case of classical reading where one has to accurately read every single base). It is also important to observe that our readouts can have a large number of symbol errors and still be associated with the correct nicking location, once again due to the availability of the reference sequence. This being said, when using NGS platforms we still get fewer user bits read due to the decreased storage density, but again, given that we do not require large depth this drawback is compensated in part.

With regards to the sequencing cost, which we did not report in the manuscript: All experiments are done in house using our own MiSeq instrument. We obtained roughly 1 million reads for each sequenced sample which was more than sufficient for error-free readout (**even when some reads had very low coverage since we know that the reads have to tile the reference sequence**). The cost is roughly to \$0.000003 per read. However, as the number of reads increases, the cost per read decrease drastically, for example, 1 billion reads cost roughly ~\$6000, resulting in \$0.00000006 per read.

About the non-integer number of reads: We thank the reviewer for noticing this issue. The problem seems to be an IGV artifact due to autoscaling and is *now fixed in the figure*.

About the difference between the coverages, a few explanations are possible: First, as we can see, shorter reads almost always have lower coverage numbers. This may be due to the fact that short reads do not get amplified appropriately or due to the fact that removing adapters from shorter reads may be more challenging. Second, the number of nicking sites and the distance between two consecutive nicking sites as well as the sequence composition might all jointly affect the number of reads that we see. **Once again, we do not need to worry about this issue as long as we have at least one read per fragment for most of the fragments, given the presence of the reference. On the theoretical side, we recently proved that the mixing scheme for registers reported in the SI retains the same rate even if one imposes strict lower bounds on the length of the reads. We termed the scheme “runlength-constrained group testing” and will report the results in a companion paper. The analysis of the new group testing scheme is highly mathematical and not suitable for publication in Nature Communications.**

“It is obvious that solid state nanopore sequencing may result in an improved reading of nicks, but if this is cost effective and scalable is still speculative. The authors might want to keep the theoretical work as part of the manuscript (preferably supp. Info), but state clearly that the combination of solid state nanopore sequencing and the proposed technology lie in the future. “

We agree with the reviewer but still believe that there is both sufficient experimental and computational evidence that nanopores are capable of sensing topological variations. Our companion paper [22] experimentally confirms this point. Still, in the revised manuscript we will clearly explain that this technology is not in a ready-to-use form yet.

“Conclusions: The following sentence is highly problematic:

“It also allows for cost-efficient scaling as a. long registers and mixtures of orthogonal registers may be nicked simultaneously; b. most uncompressed data files do not contain all possible 10-mers or compositions of orthogonal k-mers; c. genomic DNA and PfAgo, as the writing tool, are readily available...”

“a) It should read „With future developments, it may allow for cost-efficient scaling....“.

We agree and changed the sentence accordingly.

“b) Why would the authors want to store uncompressed data? The experiments were performed on compressed data.”

There are reasons to store uncompressed data in classical synthetic DNA storage systems since even single error in the compressed format can lead to catastrophic error propagation. But in our case, errors are not an issue and the above claim included an

unfortunate typo. Note that both compressed and uncompressed datasets may not contain all possible 10-mers.

We corrected the typo and wrote “most compressed data files...” instead.

“c) As discussed above, the cost of genomic DNA at sufficient quality is an important factor and is not "readily available."

We revised the statement from “readily available” to “easily extracted and used in standard laboratory settings”.

“The last sentence of the manuscript is not based on any data or discussion shown, and should be written to indicate the speculative nature of the content. (e.g. „This storage system might....)”

We changed the statement as follows: “With further improvements and optimization...”

Reviewers' Comments:

Reviewer #1:

Remarks to the Author:

The revised version of the manuscript has greatly improved in readability, the structure is now simpler and clearer, even though the general organisation and format is still problematic as figures in the main text and in supplementary material are not cited in order, some supplementary figures, tables and sections are not cited at all in the main text. This is, as I mentioned during the first revision, also the result of the large amount of data and ideas that the authors are trying to fit in a single paper. The authors should fix these remaining issues and take the occasion to better integrate data and text between the main manuscript and the supplementary material. Nonetheless, the manuscript is now more suitable for publication.

Minors:

- The platform is able to encode and decode different registers in parallel. Can the authors better discuss the accuracy of the parallel reading and how re-ordering of information is efficiently achieved?
- In simulations, it is still not clear if models of the nicked DNA were chemically saturated after removal of the backbone phosphate.
- Labels for all the graphs are way too small to be easily read.

Reviewer #2:

Remarks to the Author:

The authors have now significantly improved the quality of the manuscript. Issues of cost and sequencing coverage remain, but are less important as the scope of the manuscript has been changed and is now gives a better scientific discussion of the approach.

In my understanding, in the presented proof of concept, the authors have stored 1.3 kB of compressed digital information by nicking sections of natural DNA in 1024 well plate, by performing 1024 individual nicking reactions, and by NGS have read 80 bits of this information (8 of the the 1024 wells). If this is true, this information should be added to the abstract of the manuscript. I don't believe that this takes from the novelty or future prospects of the technology, but that it is important to give the scope of the experiments performed in the abstract.

In combination with this information, there are two shortcomings of the manuscript that are addressed in the letter to the referees, and not in the manuscript. If these points are discussed in detail, the manuscript gives a good description of the state of the art and prospects of the proposed approach.

1.) Sequencing: obviously it is currently cost and time consuming to read the written data, as the authors only had time and resources to read < 1% of the data stored. This limitation should be made clearer in the manuscript. Also, in Figure 1, from the 1024 well pool, only one arrow should be coming out of the pool, as one ssDNA pool and library is required per well.

2.) Amount of data stored: please give the amount of actual bits stored in the 1024 well plate (1024 x 10 bits = 1.3 kB ?). It is OK to keep the data on the uncompressed data, but data volume after compressions is also needed in the manuscript text. (otherwise I will come up with a storage scheme to store the compressed version of a 100 MB fully black bitmap image in a single 100 nt DNA molecule).

Added note, I don't understand the concept of the "companion paper", ref 15, which is from different authors, and can not be read online, but is from a conference?

Point by point response to the referees:

This document details our point-by-point response to all issues raised by the referees. We once again thank the editor and the reviewers for their timely and useful comments.

Our response is provided in red.

Reviewer #1 (Remarks to the Author):

“The revised version of the manuscript has greatly improved in readability, the structure is now simpler and clearer, even though the general organisation and format is still problematic as figures in the main text and in supplementary material are not cited in order, some supplementary figures, tables and sections are not cited at all in the main text. This is, as I mentioned during the first revision, also the result of the large amount of data and ideas that the authors are trying to fit in a single paper. The authors should fix these remaining issues and take the occasion to better integrate data and text between the main manuscript and the supplementary material. Nonetheless, the manuscript is now more suitable for publication.”

We have addressed the above concerns by adding references in the main manuscript to the various results in the Supplementary Information. We also made sure that all figures/tables are cited and that they appear in order.

Minors:

- “The platform is able to encode and decode different registers in parallel. Can the authors better discuss the accuracy of the parallel reading and how re-ordering of information is efficiently achieved?”

The reviewer is correct that we indeed can record information in parallel using multiple different registers (termed “orthogonal registers”). The registers were selected based on sequence similarity (requiring of them to be what we call “orthogonal”, i.e., at large edit distance from each other). Since the nicking positions were also carefully selected the fragments obtained after denaturation all align to their corresponding registers and locations without errors. Furthermore, the order of readout of the orthogonal registers is dictated by the order of their corresponding native DNA content within the *E. coli* genome. All these details are discussed in the main manuscript, under the **Results header as well as part of the **DNA register and nicking site selection** and in the Supplementary Information, under B.6. Approaches to increase the information density.**

- “In simulations, it is still not clear if models of the nicked DNA were chemically saturated after removal of the backbone phosphate”.

The reviewer is correct, and we are thankful to her/him for bringing this issue up. Indeed, the nicked DNA was chemically saturated with an H atom. We have now added this information to the Methods section.

- "Labels for all the graphs are way too small to be easily read."

We agree with the reviewer and hence increased the size of the labels.

Reviewer #2 (Remarks to the Author):

"The authors have now significantly improved the quality of the manuscript. Issues of cost and sequencing coverage remain, but are less important as the scope of the manuscript has been changed and is now gives a better scientific discussion of the approach.

In my understanding, in the presented proof of concept, the authors have stored 1.3 kB of compressed digital information by nicking sections of natural DNA in 1024 well plate, by performing 1024 individual nicking reactions, and by NGS have read 80 bits of this information (8 of the the 1024 wells). If this is true, this information should be added to the abstract of the manuscript. I don't believe that this takes from the novelty or future prospects of the technology, but that it is important to give the scope of the experiments performed in the abstract."

We thank the reviewer for the suggestions, but the point made is not correct. As mentioned, both in the main text and SI, the pool of 1024 registers with different nicking patterns does not represent the encoded data itself, but a pool of all possible 10-bit words that are subsequently used akin to Lego building blocks to encode the actual message. The actual message is stored in a larger number of well plates and we have actually stored 14.4 kB of data. We did not include this information in the abstract due to space limitations, but the description of the files is available in the main text as well as in the Supplement.

"In combination with this information, there are two shortcomings of the manuscript that are addressed in the letter to the referees, and not in the manuscript. If these points are discussed in detail, the manuscript gives a good description of the state of the art and prospects of the proposed approach.

1.) Sequencing: obviously it is currently cost and time consuming to read the written data, as the authors only had time and resources to read < 1% of the data stored. This limitation should be made clearer in the manuscript. Also, in Figure 1, from the 1024 well pool, only one arrow should be coming out of the pool, as one ssDNA pool and library is required per well.

In our case we need to sequence all intra-nick DNA fragments to retrieve the positions of the nicks which means that the yield is reduced roughly 60-fold (see the explanation in the Results section entitled **The NGS sequencing process**). There, we outlined that we need to sequence 450 base pairs to retrieve roughly 15 bits. Nevertheless, there we also

observe once again that our sequencing process is 100% accurate since we have references and that we can actually significantly save in terms of coverage as it can be really small – we know where we expect to see the nicked fragments start and end. This point was discussed in the Supplement as well.

Clearly, parallel sequencing of orthogonal registers is not a problem at all, as was described in the experimental result of multi-register recording under section B.6, Supplementary Information, **approaches to increase the information density**.

Also, we disagree that only one arrow should originate from the register pools as we can mix orthogonal registers as well as contents of different wells that contain orthogonal registers for joint sequencing.

“2.) Amount of data stored: please give the amount of actual bits stored in the 1024 well plate (1024 x 10 bits = 1.3 kB ?). It is OK to keep the data on the uncompressed data, but data volume after compressions is also needed in the manuscript text. (otherwise I will come up with a storage scheme to store the compressed version of a 100 MB fully black bitmap image in a single 100 nt DNA molecule). “

As already described in the previous response, the 14.4 KB number reported represents the size of the compressed data. We added the actual file sizes before compression into the revised manuscript to avoid confusion. We were under the impression after the first round of revision that the reviewer only wanted us to report the actual volume stored which is that of the compressed file size (compression is performed before recording).

“Added note, I don't understand the concept of the "companion paper", ref 15, which is from different authors, and cannot be read online, but is from a conference?”

The paper was co-authored by two of the authors of this paper. Nevertheless, we fixed the references and renamed the work as “follow-up paper”. Also, we added one more reference of a follow-up to our work that appeared recently and demonstrates the in-memory parallel sorting capabilities of our storage platform.